# Continual Vision-Language Representaion Learning with Off-Diagonal Information

## Abstract

Multimodal pre-trained methods with a contrastive learning framework (like CLIP) have recently achieved consistent advantages on various cross-model downstream tasks. However, they usually require a large amount of image-text samples and a vast computing budget for training, which makes the re-training process expensive while the training data is collected continuously (the phenomenon is widespread in real scenarios). In this paper, we discuss the feasibility of continuously training CLIP models based on discrete streaming data. We find that the multimodal retrieval performance of the CLIP in a continual training setting is significantly lower than that in a joint training setting. We name this phenomenon **Cognitive Disorder(CD)**. By tracking the directional changes of the representation vectors in the continuously updated CLIP model, we explore and summarize the spatial variation of the modal encoders within the CLIP: **Intra-modal Rotation** and **Inter-modal Deviation**. Intra-modal Rotation means that the vision and language representation space in the CLIP is rotating greatly around the center of a high-dimensional unit sphere during continual training, accompanied by a relatively small change in the topology of the representation space. Inter-modal deviation happens when the vision and language's intra-modal rotation is unsynchronized. Moreover, we empirically and theoretically demonstrate how intra-modal rotation and inter-modal deviation lead to CD. In order to alleviate CD in continual CLIP training, we propose a new continual training framework **Mod-X**: **M**aintain **o**ff-**d**iagonal information-matri**X**. By selectively aligning the off-diagonal information distribution of contrastive matrixes, the Mod-X helps the model not only better fits the newly trained data domain but also maintains the multimodal cognitive ability on the old data domain during the continual large-scale training (Section 5).

## 1 Introduction

Recently, multimodal pre-trained models such as CLIP Radford et al. (2021) have attracted much attention. By utilizing these pre-trained models, many works have achieved new progress in downstream tasks such as classification Zhang et al. (2020); Wei et al. (2022); Lee et al. (2022), semantic segmentation Xie et al. (2021b); Wang et al. (2021b), object detection Xie et al. (2021a); Wang et al. (2022a), speech recognition Baevski et al. (2020), etc. Although the CLIP model has strong generalization in open-world data, as mentioned in CLIP paper Radford et al. (2021), the ability to match image-text samples that are not in its training data distribution is still weak. The natural idea to alleviate this problem is to scale the training data to cover different data domains. However, it is infeasible to train infinite data with limited hardware at once.

In this paper, trying to break this non-iterability, we explore the feasibility of continuously training the CLIP model through streaming data, a training paradigm that follows Continual Learning (CL) McCloskey & Cohen (1989). To simulate continual CLIP training, we randomly and evenly divide the training data (joint-dataset) into multiple sub-datasets and train the CLIP sequentially using these sub-datasets. For comparison with continual training, we train a CLIP additionally from scratch using the joint-dataset, which is named joint training, as the upper bound on the performance of the continuously trained model.

Traditional supervised continual learning has been proven to suffer from catastrophic forgetting Rebuffi et al. (2017); Kirkpatrick et al. (2017). The model's performance on old tasks drops significantly

as training phases rise. Recently, some work Ni et al. (2021b); Hu et al. (2021) has validated that self-supervised models like SimCLR Chen et al. (2020) and BarlowTwins Zbontar et al. (2021) do not suffer from severe catastrophic forgetting during continual training. Some works Madaan et al. (2021); Thai et al. (2021) conjecture that the reason is that the contrastive loss is not directly affected by the supervised signal, and the self-supervised framework does not have a SoftMax function to amplify the influence of labels.

However, the performance of CLIP with a continual training setting overturns this hypothesis. There is a significant degradation of multimodal retrieval results with continual training compared with joint training (in Section 3 and 5). We name this phenomenon **Cognitive Disorder(CD)**. Due to the vision and language encoders within the CLIP normalizing the representation to a unit vector through a dimension-based $L_2$ norm, which limits the diffusion of representation vectors length, we try to analyze the representation space variation of modal extractors from a spatial geometry perspective. By tracking the directional changes of the representation vectors in the continuously updated CLIP model (in Section 3), we explore and summarize the spatial variation of the modal encoders within the CLIP: the **Intra-modal Rotation** and **Inter-modal Deviation**. The intra-modal rotation refers to the representation space of the single-modal feature extractors (vision and language) within the CLIP that rotates around the center of the high-dimensional sphere, accompanied by a slow topology change during the continual CLIP training. The inter-modal deviation refers to the cognitive deviation of different modal extractors (vision and language) to the same entities during continuous training. Moreover, we empirically and theoretically demonstrate how intra-modal rotation and inter-modal deviation lead to cognitive disorder (in Section 3).

To alleviate this cognitive disorder in continual CLIP training, we propose a simple yet effective framework Mod-X: **M**aintain **o**ff-**d**iagonal information-matri**X**. Unlike contrastive loss Oord et al. (2018) only focuses on the proportion of positive and negative sample pairs. The Mod-X framework pays more attention to the distribution of off-diagonal information in the contrastive matrix. The similarity distribution on the off-diagonal illustrates the model's cognition of all entities on current data. By selectively aligning the off-diagonal information distribution of the contrastive matrixes constructed by the current and past models based on the recent training sample, Mod-X helps the model preserve the correct cognition of various old entities while fitting the current vision-language data during continual training. The evaluations in Experiments 5 with different scale and scope datasets show that our Mod-X framework helps the model not only better fits the newly trained data domain (in Section 5.3) but also maintains the multimodal cognitive ability between the current model and old model on the old data domain during the continual large-scale training (in Section 5.4). More technological details and evaluations have been shown in Section 4 and Section 5. In summary, our contributions are as follows:

- We discuss the feasibility of training the CLIP model continuously through streaming data. Empirical experiments demonstrate that continual CLIP training leads to persistent performance degrades on multimodal retrieval. We name this Cognitive Disorder.

- By introducing a series of tools to track the directional changes of the representation vectors in the continuously updated CLIP model, we explore and summarize the spatial variation of the modal encoders within the CLIP: 1) The Intra-modal Rotation 2) The Inter-modal Deviation. Furthermore, we empirically and theoretically demonstrate how intra-modal rotation and inter-modal deviation lead to cognitive disorder (in Section 3).

- We propose a simple yet effective continual CLIP training framework **Mod-X** that alleviates CLIP's cognitive disorder by selectively aligning off-diagonal information in contrastive matrixes between the past and current models in continual training.

## 2 RELATED WORK

**Continual Learning.** Continual learning (CL) Thrun (1995), or incremental learning, is mainly focused on supervised tasks. In addition to the vision-based tasks De Lange et al. (2021); Kj et al. (2021); Cha et al. (2021); Ahn et al. (2021), some works discussing language-based tasks Biesialska et al. (2020); Sun et al. (2019). We can summarize the existing continual learning methods into three categories: regularization Kirkpatrick et al. (2017); Ahn et al. (2019); Ni et al. (2021a), replay Rebuffi et al. (2017); Rolnick et al. (2019); Wang et al. (2021a), and architecture Thai et al. (2021); Ni et al. (2021b); Hu et al. (2021); Madaan et al. (2021). In unsupervised and self-supervised based on a

single modal, the latest work Thai et al. (2021); Ni et al. (2021b); Hu et al. (2021); Madaan et al. (2021) has drawn some conclusions different from those of supervised. However, only a few works Srinivasan et al. (2022); Fan et al. (2022) focus on incremental multimodal tasks learning. Because of the cooperation between different models, continual multimodal pre-training shows different performance and problems from single modal continual training.

**Visual-Language Representational Learning.** Vision-language representation learning based on contrastive loss Oord et al. (2018) such as CLIP Radford et al. (2021) has attracted a lot of attention in various fields Radford et al. (2021); Li et al. (2021); Andonian et al. (2022); Mahabadi et al. (2022). And the pre-trained model perform surprisingly well on downstream tasks Shu et al. (2022); Wang et al. (2022b); Chowdhury et al. (2022). At the same time, the large-scale image-text datasets e.g., Laion400M Schuhmann et al. (2021) and Conceptual Captions Sharma et al. (2018) has played a key role in multimodal pre-training. Although large-scale open-world datasets contain various samples, pre-trained model still cannot perfectly match image-text sample pairs that are not in its training data domain Radford et al. (2021).

## 3 COGNITIVE DISORDER IN CONTINUAL CLIP TRAINING

This section discuss the performance of the $CLIP_{ct}$ model, which trains CLIP continually without other operations. We name the decline in $CLIP_{ct}$ performance as **Cognitive Disorder**. At the same time, the spatial variation of the modal encoders within the $CLIP_{ct}$ are explored and summarized: 1). Intra-modal Rotation 2). Inter-modal Deviation. Finally, we empirically and theoretically demonstrate how intra-modal rotation and inter-modal deviation lead to cognitive disorder.

**Exploration Setup.** To ensure the controllability of the exploration, we train a $CLIP_0$ model from scratch on the COCO dataset Lin et al. (2014) based on the OpenAI source code OpenAI and use it as the initial(start) of continual CLIP training. After that, we divide the Flickr30K dataset Young et al. (2014) into five sub-datasets $\{D_1, D_2, D_3, D_4, D_5\}$ uniformly and randomly to simulate streaming data. Then we train the $CLIP_0$ based on this sub-datasets sequentially, and no other data is used when training on each sub-dataset. After finishing five training phases, we obtain the model $CLIP_5$. For comparison with $CLIP_5$, we train a $CLIP_{jt}$ from scratch using joint dataset COCO and Flickr30K as the upper bound of the $CLIP_{ct}$. The hyper-parameters for all training phases are kept the same, and detailed settings of CLIP can be seen in Appendix 7.4.

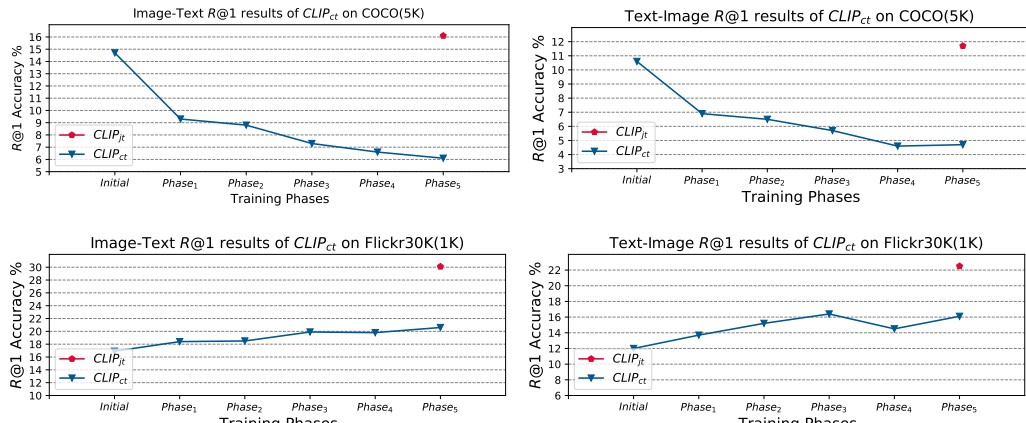

Figure 1: The multimodal retrieval $R@1$ results of $CLIP_t$ ($0 \leq t \leq 5$) on test sets COCO (5K) and Flickr30k (1K). The two sub-figures on the left show the Image-Text retrieval $R@1$ performance of $CLIP_t$ on the continual training phase $t$. The rights show the Text-Image $R@1$ results. The pentagon points ($CLIP_{jt}$) show the results of the CLIP under joint training, which is an upper bound for continual CLIP training ($CLIP_{ct}$).

### 3.1 THE PERFORMANCE OF CONTINUAL CLIP TRAINING

We show the $R@1$ retrieval results of $CLIP_t$ ($0 \leq t \leq 5$) on the 5K test set of COCO (COCO(5K)) and 1K test set of Flickr30k (Flickr30K(1K)) in Figure 1. By comparing the multimodal retrieval

performances of the CLIP$_0$ (initial phase) and CLIP$_{jt}$ on Flickr30K(1K), we can find that the retrieval performance of CLIP$_{jt}$ is significantly better than that of CLIP$_0$, which is not trained on Flickr30k. This phenomenon shows that the performance of the CLIP model is affected by the training data domain, which is consistent with the results of the paper Radford et al. (2021). Besides this, it can be clearly seen that the multimodal retrieval performance of the CLIP$_{ct}$ on the COCO(5K) has continued to decline with the rise of training phases. The final Image-Text R@1 result of CLIP$_5$ on COCO(5K) plummeted from the initial 14.7% to 6.1%, and the Text-Image results dropped from 10.6% to 4.7%. The gap with CLIP$_{jt}$ reached 10.0% and 7.0%. On the other hand, CLIP$_{ct}$ exhibits a slow and erratic increase in multimodal retrieval $R@1$ results on the test set Flickr30K(1K). Although the results between CLIP$_{ct}$ and CLIP$_{jt}$ on the Image-Text $R@1$ has been narrowed from the original 13.2% to 9.5% while the Text-Image $R@1$ of CLIP$_{ct}$ has increased from 12.0% to 16.1%, the gap between CLIP$_5$ and CLIP$_{jt}$ is still great. We name this phenomenon **Cognitive Disorder (CD)**.

## 3.2    THE REASONS FOR COGNITIVE DISORDER

In CLIP, the vision and language encoders normalize the final representation vector to a unit vector of length 1 using a dimension-based $L_2$ norm. This design makes the representation space in vision and language encoders form a high-dimensional unit sphere. Based on this fact, we ignore the influence of the representation vectors' length and track their direction changes.

### 3.2.1    THE INTRA-MODAL ROTATION

Firstly, we analyze the directional changes of the representation vectors of vision and language extractors in continual CLIP training. Taking the visual representation space as an example, we use the visual encoder $E_i^V$ in CLIP$_i$ to extract the image representations of the test set COCO(5K) and obtain the vision representation vectors sets $V_i = \{V_i^0, V_i^1, V_i^2, ..., V_i^N, ..., V_i^{5K}\}$, where $i = 0, ..., 5$ stands for five different training phases. After that, we take the inner product of each pair of vectors $< V_i^a, V_i^b >$ in each vector set $V_i$ and perform $arccos$ operation to obtain their **S**elf-**A**ngle relationship **M**atrix ($SAM_i$). The $SAM_i^{(a,b)} = arccos(< V_i^a, V_i^b >)$. Any element $SAM_i^{(a,b)}$ in the $SAM_i$ matrix represents the included angle between the sample $a$ and $b$ in the vision encoder $E_i^V$. By counting the difference $\theta_{SAM}$ between the corresponding elements in two continual $SAM$ matrix $SAM_i$ and $SAM_{i+1}$ as shown in Figure 2(a), we get the following included angle change distribution table 2(b).

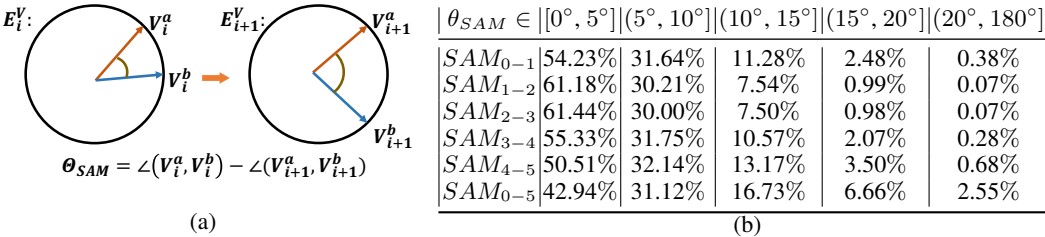

| $\theta_{SAM} \in$ | $[0°, 5°]$ | $(5°, 10°]$ | $(10°, 15°]$ | $(15°, 20°]$ | $(20°, 180°]$ |
|---|---|---|---|---|---|
| $SAM_{0-1}$ | 54.23% | 31.64% | 11.28% | 2.48% | 0.38% |
| $SAM_{1-2}$ | 61.18% | 30.21% | 7.54% | 0.99% | 0.07% |
| $SAM_{2-3}$ | 61.44% | 30.00% | 7.50% | 0.98% | 0.07% |
| $SAM_{3-4}$ | 55.33% | 31.75% | 10.57% | 2.07% | 0.28% |
| $SAM_{4-5}$ | 50.51% | 32.14% | 13.17% | 3.50% | 0.68% |
| $SAM_{0-5}$ | 42.94% | 31.12% | 16.73% | 6.66% | 2.55% |

(a)                                                                    (b)

Figure 2: The sub-figure on the left shows a schematic diagram of computing $\theta_{SAM}$. The table on the right shows the distribution of the change of the included angle between any two samples in different training phases' vision representation space. And $SAM_{i-j} = |SAM_i - SAM_j|$.

From the table 2(b), we can find that 80% of the angle changes between any two vision representation vectors in the vision representation space are between 0 and 10 degrees in the process of continual CLIP training, while only 20% are above 10 degrees. Moreover, less than 1% of the angle changes are above 20 degrees. Those angle changes between 15-20 degrees also only account for about 5% of all image pairs. Therefore, we can conclude that **the topology of the visual representation of the CLIP$_{ct}$ changes slowly during the continual CLIP training.** In Appendix 7.9, we discuss this conjecture by comparing the representation quality of vision encoders.

In addition to discussing the change in the included angle between sample pairs in the visual representation space, by taking the inner product of the same sample's vision representation vector from different training phases' vision encoder $E_i^V$, we use the $arccos$ operation to compute the rotation angles $\theta_{RAM}$ of each test sample in vision encoder $E_i^V$ and $E_j^V$ and get the **R**otation **A**ngle

**M**atrix RAM$_{(i,j)}$. The $RAM_{(i,j)}^a = arccos(< V_i^a, V_j^a >)$, where the $a$ is the label of sample. The schematic diagram can be seen in 3(a). By counting the distribution of rotation angles, we get the following rotation angle distribution table 3(b).

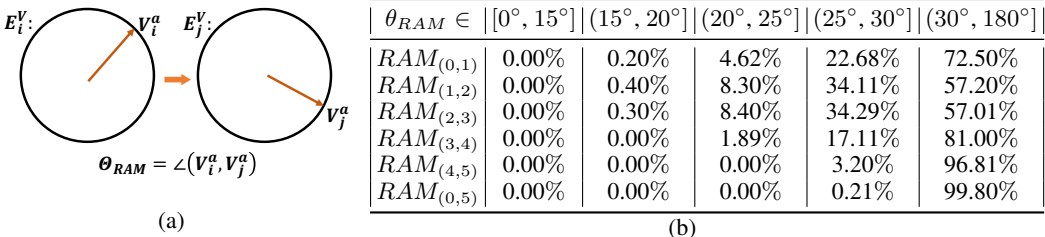

| $\theta_{RAM} \in$ | $[0°, 15°]$ | $(15°, 20°]$ | $(20°, 25°]$ | $(25°, 30°]$ | $(30°, 180°]$ |
|---|---|---|---|---|---|
| $RAM_{(0,1)}$ | 0.00% | 0.20% | 4.62% | 22.68% | 72.50% |
| $RAM_{(1,2)}$ | 0.00% | 0.40% | 8.30% | 34.11% | 57.20% |
| $RAM_{(2,3)}$ | 0.00% | 0.30% | 8.40% | 34.29% | 57.01% |
| $RAM_{(3,4)}$ | 0.00% | 0.00% | 1.89% | 17.11% | 81.00% |
| $RAM_{(4,5)}$ | 0.00% | 0.00% | 0.00% | 3.20% | 96.81% |
| $RAM_{(0,5)}$ | 0.00% | 0.00% | 0.00% | 0.21% | 99.80% |

(a)        (b)

Figure 3: The sub-figure on the left shows a schematic diagram of computing $\theta_{RAM}$. The table on the right shows the rotation angle distribution of the same samples in different training phases.

By observing the table 3(b), we can find that the direction of the same sample in the visual representation space of different training phases has changed greatly. Only less than 0.4% samples are rotated within 20 degrees in the continual CLIP training, while the samples rotated within 20-25 degrees are at most less than 9%, and the samples of 25 degrees and above account for more than 90%. We speculate that **the vision representation space of CLIP$_{ct}$ has undergone a large rotation around the high-dimensional sphere center during the continual training.** After analyzing the language representation space, we reach the same conclusion as the vision representation space. Detailed SAM and RAM distribution of language encoders can be viewed in Appendix 7.2.

According to our analysis of the geometric changes of the single-modal encoder's representation space during continual CLIP training, we conclude that: **During the continual CLIP training, the representation space in the CLIP$_{ct}$ is significantly rotated. The topological structure of the representation space is slightly rotated compared with the rotation of the whole representation space.** We name this phenomenon **Intra-modal Rotation**.

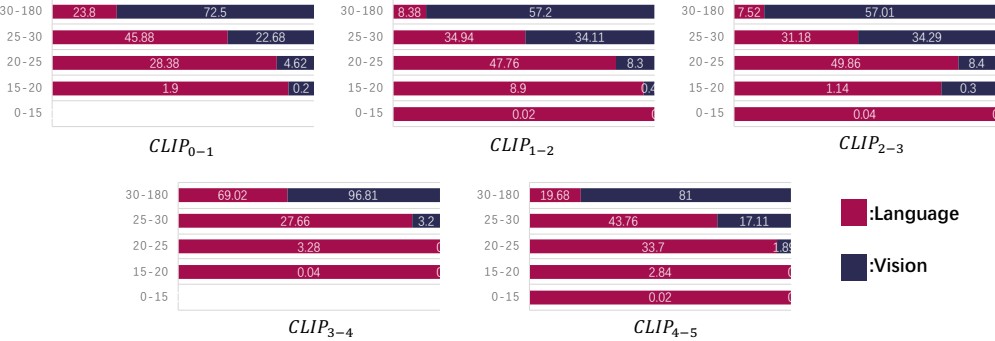

Figure 4: The comparison of the rotation distributions of the representation spaces of the vision and language extractors during continual CLIP training. CLIP$_{i-j}$ refers to the CLIP's continual training from training phase $i$ to $j$.

### 3.2.2 THE INTER-MODAL DEVIATION

Although the topology of the single-modal representation space changes during continual training, this slight rotation should not be the main reason for the degradation of CLIP's multimodal retrieval performance. To this end, we conduct a thought experiment: it is known that the representation spaces of vision and language encoders exhibit significant spatial rotations during continual training. Now we assume that the topology of the single-modal representation space is completely fixed during continual training. Therefore, if the CLIP$_{ct}$'s performance on multimodal retrieval does not degrade during continual training, the rotations of the two encoders' representation spaces should be **synchronized**. However, the fact is the opposite. So we think **there is a deviation between the rotation of the vision and language representation spaces**. Based on this suppose, we compare the rotation distributions of vision encoder (Figure 3(b)) and language encoder (Table 7(b)) and

draw the rotation distribution comparison diagram (Figure 4). The values under the same color represent the proportion of test samples to total samples in each rotation angle interval of the same modality. Comparing the difference in the distribution of rotation angles of the vision and language encoders, we can see that the space rotations of the two encoders are very different in the continual training. The rotation of language representation space is mostly concentrated between 20-30 degrees, while the vision's rotations are mostly between 30-180 degrees. This shows that the rotation of the representation space of the two modal extractors within $CLIP_{ct}$ is not synchronized during the continual training, which verifies our previous inference: **The unsynchronized rotation of the vision and language representation spaces leads to cognitive deviations between the CLIP's modal encoders (vision and language).** We name this phenomenon **Inter-modal Deviation**.

### 3.2.3 INTRA-MODAL ROTATION AND INTER-MODAL DEVIATION LEAD TO COGNITIVE DISORDER

Based on the above exploration, we can conclude that intra-modal rotation and inter-modal deviation play a key role in $CLIP_{ct}$'s cognitive disorder. However, how do they cause the model to misalign the old sample's vision and language representation? We show a schematic here to illustrate this. As shown in Figure 5, the $\alpha$ is vision representation and $\beta$ is language representation. The $a,b$ denote different image-text samples. For the convenience of illustration, we set the unsynchronous rotation of the two modal spaces as the visual modal's static and the language's relative rotation. When intra-modal rotation happens 5(a), $\beta_a$ in training phase $t + 1$ is rotated to $\beta'_a$, the modal similarity between $a$ and $b$ shift from $(\beta_a^T \alpha_a > \beta_a^T \alpha_b)$ to $(\beta'^T_a \alpha_a < \beta'^T_a \alpha_b)$, which break the alignment of the current model to old sample $a$. The superscript $T$ is a transpose operation that is often used for matrix multiplication. When inter-modal deviation happens 5(b), the relative rotation of the representation space breaks the original modal alignment of sample $a$ which makes the $(\beta_a^T \alpha_b > \beta_a^T \alpha_a)$. Because of this, the performance of $CLIP_{ct}$ drops significantly during continual training. Detailed mathematical derivations can be found in Appendix 7.1.

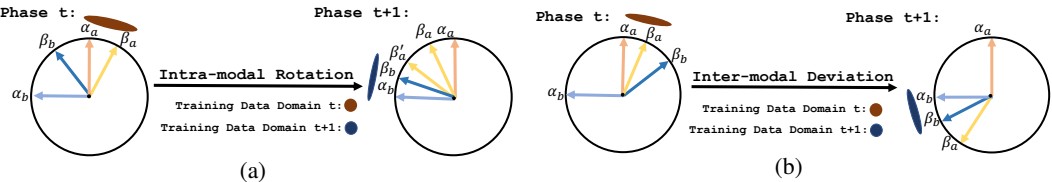

(a)                                              (b)

Figure 5: The Schematic illustration of cognitive disorder caused by intra-modal rotation and inter-modal deviation.

## 4 ALLEVIATING COGNITIVE DISORDER

### 4.1 GENERAL CONTINUAL CLIP TRAINING SETTING

Suppose we have used training dataset $D_0$ got a pre-trained model $CLIP_0$. And there is another vision-language dataset $D$. We split $D$ into $N$ sub-datasets $\{D_1, ..., D_N\}$, randomly and evenly, to simulate a stream of data and $D_t = \{(v_t^0, l_t^0), ..., (v_t^n, l_t^n)\}$ denotes the training data in the training phase $t$, where $t \in \{1, 2, ..., N\}$. Then training the model $CLIP_0$ using sub-datasets sequentially. The encoded $l_2$ normalized embeddings of vision and text is $V_t^i = E_V^t(v_t^i)$ and $L_t^i = E_L^t(l_t^i)$. When the model $CLIP_t$ is trained during the training phase $t$ using training data $D_t$, the previous sub-datasets $\{D_0, D_1, ..., D_{t-1}\}$ are no longer available. The joint training refers to training a $CLIP_{jt}$ from scratch using all data $D_{jt} = D_0 \cup D$.

### 4.2 MOD-X: MAINTAIN OFF-DIAGONAL INFORMATION-MATRIX

To alleviate cognitive disorder of the $CLIP_{ct}$ model during continual training. We produce a simple but effective new training framework: Maintain off-diagonal information-matrix (Mod-X). It helps the current CLIP retain the cognition of past samples by aligning off-diagonal information in the similarity matrix constructed by the $CLIP_{ct}$ before and after continual training based on the current

training data. The entire training framework is shown in Figure 6. The Contrastive module in Figure 6 is a traditional InfoNCE loss Baevski et al. (2020) , which inherit from CLIP Radford et al. (2021). In there, we mainly introduce our Cognition Align module.

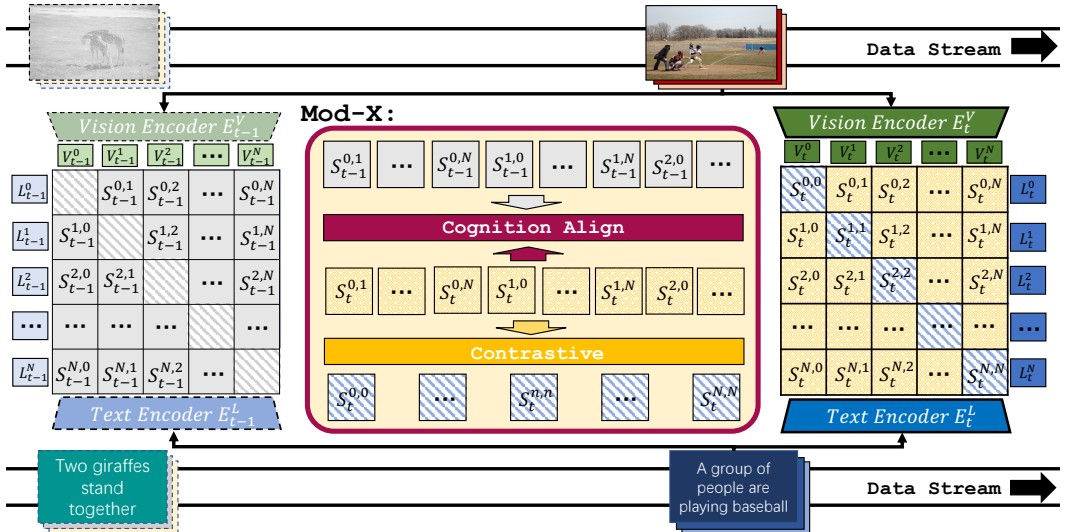

Figure 6: The Mod-X framework mainly consists of two subsections. Cognition Align helps the current model align the cognition of the old model based on current data. And Contrative helps the model fit the current training data domain. The $S^{i,j}$ means cosine similarity score of the $i$'th sample vision embedding and the $j$'th sample text embedding.

## 4.3 COGNITION ALIGN

The diagonal elements in CLIP's contrastive matrix represent the similarity of the model's understanding of the visual and language information of the current sample. The off-diagonal elements represent the similarity between the vision and language representation of the current sample and other samples or vice versa. The distribution of off-diagonal elements in the contrastive matrix represents the current model's cognition about the current training objects. So we use Cognition Align to distill the old model's "world view" of current samples to help the current model maintain the cognitive ability on past entities. Firstly, we construct contrastive matrix $M_{t-1}$ and $M_t$ using the last and current model $CLIP_{t-1}$ and $CLIP_t$ based on current sub-dataset $D_t$.

$$M_{t-1}^{i,j} = CLIP_{t-1}(D_t) = s(E_V^{t-1}(v_t^i), E_L^{t-1}(l_t^j)) \tag{1}$$

$$M_t^{i,j} = CLIP_t(D_t) = s(E_V^t(v_t^i), E_L^t(l_t^j)) \tag{2}$$

Where the $s(a,b) = (a)^T b$ is the cosine similarity function. However, the last model's cognition for current data is not totally correct. For those misunderstood sample information (diagonal elements are not the largest in the current retrieval), we use the corresponding similarity information of the current model to replace them. Thereby removing their influence on the current model during distillation.

$$M_{t-1}^{(i,:)} = M_t^{(i,:)}; \quad if \quad max(M_{t-1}^i) \neq i \tag{3}$$

After that, we align the information matrix $M^{t-1}$, which is selected by Screening, using Kullback-Leibler Divergence Csiszár (1975).

$$L_{KL}^t(M^t, M^{t-1}) = -\sum M^{t-1} ln(\frac{M^t}{M^{t-1}}) \tag{4}$$

The final training loss can be joint in $L_{Mod-X}$ and $\alpha$ is a hyper-parameter.

$$L_{Mod-X}^t = L_{InfoNCE}^t + \alpha L_{KL}^t \tag{5}$$

## 5 EXPERIMENTS

### 5.1 TRAINING DATASETS

In the experiments, we use three image-text datasets varying in scale and scope to simulate continual CLIP training. **MS COCO Captions** Lin et al. (2014): MS COCO Captions (COCO) is a widely used image caption dataset. It contains 80K training images, 30K validation images, and 5K testing images (COCO(5K)). Each image has 5 human-evaluated captions. **Flickr30K** Young et al. (2014): Flickr30K(F30K) contains 30K training images and 1K test samples (Flickr30K(1K)) collected from Flickr, together with 5 reference sentences provided by human annotators. **SBU-1M** Ordonez et al. (2011): The SBU-1M collects 1M images from Flickr with associated visually relevant captions which representing a wider variety of content styles. After downloading and preprocessing, we utilized about 0.8M image-text pairs in our experiments.

### 5.2 EXPERIMENTS SETTING

We apply our Mod-X framework to two experiment settings to illustrate that Mod-X helps the model not only better fits the newly trained data domain but also maintains the multimodal cognitive ability of the old data domain during the continual large-scale training. **Experiment A:** The Experiment A follows the setup of the exploratory experiments, pre-training the $CLIP_0$ model using the COCO dataset. And then splitting Flickr30K dataset randomly and evenly into five sub-datasets and sequentially training them to update $CLIP_0$ continually. **Experiment B:** Unlike Experiment A, which looks like fine-tuning, we try to simulate continual large-scale training in Experiment B. We train a CLIP from scratch using the joint data of COCO and Flickr30K (COCO+F30K) as the initial model $CLIP_0$ and continually train eight sub-datasets divided from the SBU-1M dataset, randomly and uniformly. Each sub-dataset contains 100K unique samples, an equal scale to the initial pre-training dataset (COCO+F30K). This experiment verifies that our framework can iterate under continual large-scale training and maintain or even improve the model's cognition of past entities. **Evaluation Details:** We use the test sets COCO(5K) and Flickr30K(1K) to evaluate the performance of model in continual training. And all the results in the tables and figures are the average of 3 runs with different seeds. Experiment details have been shown in Appendix 7.4.

### 5.3 PERFORMANCE OF MOD-X IN THE EXPERIMENT A

The Experiment A focuses on verifying that Mod-X can not only fit new data domains but also maintain a cognition of past data domains. This table shows the final multimodal retrieval performance of the different training strategies on COCO(5K) and Flickr30K(1K). The $CLIP_0$ is the initial pre-trained model based on the COCO dataset. The $CLIP_{ct}$ means training CLIP continually without any other operation. The $CLIP_{EWC}$ means using the EWC method Kirkpatrick et al. (2017), which is a typical regularization strategy in continual supervised learning. The $CLIP_{Mod-X}$ is the proposed Mod-X, and the $CLIP_{JT}$ is the joint training model using the joint datasets (COCO+F30K), which is an upper bound for continual CLIP training in Experiment A. The detailed performance of the different models at each training phase is shown in Appendix 7.5.

| Pretraining Dataset | Model | Image-Text Retrieval(%) | | | | | | Text-Image Retrieval(%) | | | | | |
| | | Flickr30K(1K) | | | COCO(5K) | | | Flickr30K(1K) | | | COCO(5K) | | |
| | | $R@1$ | $R@5$ | $R@10$ | $R@1$ | $R@5$ | $R@10$ | $R@1$ | $R@5$ | $R@10$ | $R@1$ | $R@5$ | $R@10$ |
|---|---|---|---|---|---|---|---|---|---|---|---|---|---|
| COCO | $CLIP_0$ | 16.9 | 37.0 | 46.2 | 14.7 | 34.2 | 47.0 | 12.0 | 30.0 | 41.0 | 10.6 | 29.6 | 41.0 |
| | $CLIP_{ct}$ | 20.6 | 42.8 | 56.4 | 6.2 | 17.8 | 26.1 | 16.1 | 38.5 | 50.4 | 4.7 | 14.3 | 21.8 |
| | $CLIP_{EWC}$ | 22.2 | 43.1 | 57.0 | 6.1 | 17.2 | 26.5 | 17.0 | 39.1 | 51.2 | 4.5 | 13.9 | 22.0 |
| | $CLIP_{Mod-X}$ | **27.9** | **53.4** | **64.4** | **14.5** | **34.0** | **46.1** | **20.2** | **45.0** | **57.2** | **10.1** | **26.4** | **37.4** |
| COCO+F30K | $CLIP_{jt}$ | 30.1 | 55.9 | 60.1 | 16.1 | 38.1 | 51.9 | 22.5 | 48.5 | 59.6 | 11.7 | 30.9 | 42.7 |

Table 1: The final multimodal retrieval performance of the different continual CLIP training strategies in the Experiment A.

From the results in the Table 1, it is clear that our method $CLIP_{Mod-X}$ maintains its multimodal retrieval results on COCO(5K) after completing continual training on Flickr30K. The gap between

$CLIP_0$ and $CLIP_{Mod-X}$ is just 0.2% points in image-text retrieval and 0.5% points in text-image retrieval on COCO(5K). At the same time, the retrieval results of the $CLIP_{Mod-X}$ on the test set Flickr30K(1K) are also affected by the training domain and have a significant increase. The $R@1$ performance of the $CLIP_{Mod-X}$ in image-text retrieval rise from 16.9% (in $CLIP_0$) to 27.9%. And the $R@1$ results in text-image retrieval increase from 12.0% (in $CLIP_0$) to 20.2%. The performance gap between $CLIP_{Mod-X}$ and $CLIP_{jt}$ on the Flickr30K is only at most 2.3% points. Conversely, due to the model's cognitive disorder in continual training, the performance of $CLIP_{ct}$ on COCO(5K) drops significantly. In addition, although the performance of $CLIP_{ct}$ on Flickr30K(1K) has improved, it is still far from the upper bound $CLIP_{jt}$. The EWC, as a typical regularization strategy in continual learning, selectively updates the model by evaluating the importance of each parameter of the model. From the above experimental results, although $CLIP_{EWC}$ improves the accuracy of continual CLIP training on Flickr30K(1K), it does not preserve the model's understanding in past samples (COCO(5K)). According to the above comparisons, we can conclude that our Mod-X framework can not only maintain the cognitive ability on past samples during continual CLIP learning but also improve the model's fitting ability to the current training data domain.

## 5.4 PERFORMANCE OF MOD-X IN THE EXPERIMENT B

Unlike Experiment A, which looks like fine-tuning, Experiment B focuses on verifying that our framework Mod-X can maintain or even improve the model's understanding of past entities when iterating under continual large-scale training. Table 2 shows the final multimodal retrieval performance of the different continual CLIP training strategies in Experiment B. The detailed $R@1$ results in each training phase can be seen in Appendix 7.6.

| Pretraining Dataset | Model | Image-Text Retrieval(%) | | | | | | Text-Image Retrieval(%) | | | | | |
| --- | --- | --- | --- | --- | --- | --- | --- | --- | --- | --- | --- | --- | --- |
| | | Flickr30K(1K) | | | COCO(5K) | | | Flickr30K(1K) | | | COCO(5K) | | |
| | | $R@1$ | $R@5$ | $R@10$ | $R@1$ | $R@5$ | $R@10$ | $R@1$ | $R@5$ | $R@10$ | $R@1$ | $R@5$ | $R@10$ |
| COCO+F30K | $CLIP_0$ | 33.0 | 56.5 | 67.3 | 17.7 | 40.7 | 52.9 | 23.4 | 48.2 | 59.1 | 12.9 | 32.8 | 44.9 |
| | $CLIP_{ct}$ | 5.4 | 16.0 | 23.3 | 3.0 | 9.0 | 14.0 | 3.6 | 12.0 | 18.6 | 2.3 | 7.3 | 11.4 |
| | $CLIP_{EWC}$ | 6.1 | 18.2 | 33.5 | 3.1 | 11.7 | 18.2 | 3.3 | 10.4 | 16.2 | 2.1 | 5.4 | 10.0 |
| | $CLIP_{Mod-X}$ | **37.3** | **62.6** | **72.1** | **19.5** | **42.8** | **55.7** | **26.0** | **51.9** | **62.6** | **13.4** | **33.6** | **45.3** |
| Joint Datasets | $CLIP_{jt}$ | 39.3 | 71.5 | 81.1 | 25.1 | 51.6 | 64.3 | 27.8 | 54.9 | 66.5 | 15.9 | 38.6 | 51.3 |

Table 2: The final multimoal retrieval performance of the different continual CLIP training strategies in the Experiment B.

Compared to the initial pre-trained model $CLIP_0$, our $CLIP_{Mod-X}$ not only does not show a significantly drop such as $CLIP_{ct}$ in multimodal retrieval performance on COCO(5K) and Flickr30K(1K) but it also shows a slight improvement after continual training in SBU-1M. The image-text $R@1$ results on Flickr30K(1K) increase from 33.0% to 37.3% and the accuracy on COCO improved to 19.5% from 17.7%. The performance of text-image $R@1$ is the same as that of image-text. The accuracy increase from 23.4% to 26.0% on Flickr30K(1K) and 12.9% to 13.4% on COCO(5K). The gap between continual CLIP training and joint training has been somewhat narrowed. Conversely, the ability of cognitive alignment in $CLIP_{ct}$ and $CLIP_{EWC}$ are lost in the large-scale data. This results in verifying that our framework Mod-X can iterate under continual large-scale training and can maintain or even improve the model's cognitive ability of past entity concepts.

## 6 CONCLUSION

In this paper, We explore the feasibility of training the CLIP model continuously through streaming data and name its performance decline in multimodal retrieval as Cognitive Disorder(CD). Then, By tracking the directional changes of the representation vectors in the continuously updated CLIP model, we explore and summarize the spatial variation of the modal encoders within the CLIP: Intra-modal Rotation and Inter-modal Deviation. Moreover, we mathematically demonstrate how intra-modal rotation and inter-modal deviation lead to CD. To alleviate the cognitive disorder of the continual CLIP training, we propose a simple yet effective continual learning framework Mod-X: **M**aintain **o**ff-**d**iagonal information-matri**X**. The results in experiments (5.3 and 5.4) demonstrate the effectiveness of our framework.

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

# 7 APPENDIX

## 7.1 THE THEORETICAL DEMONSTRATE THAT INTER-MODAL DEVIATION AND INTRA-MODAL ROTATION LEAD TO COGNITIVE DISORDER

Inter-modal Deviation and Intra-modal Rotation can influence the CLIP's sample similarity matrix, but this does not necessarily lead to errors in multimodal retrieval results. Unless the similarity of the visual language representation of the model for the same sample is smaller than that between different samples. In there, we abstract this problem and give the theoretical conditions that the Intra-modal Rotation and Inter-modal Deviation leads to the cognitive disorder.

There has $N$ image-text pairs $\{(\alpha_1,\beta_1),(\alpha_2,\beta_2),(\alpha_3,\beta_3),...,(\alpha_i,\beta_i),...,(\alpha_N,\beta_N)\} \in R^{W \times W}$. Through function $\mathcal{M}(\alpha)$ and $\mathcal{Q}(\beta)$, $\mathcal{M} \neq \mathcal{Q}$, the Euclidean space A and B of images and texts are formed.

$$
\begin{aligned}
A &= span\{\mathcal{M}(\alpha_1), \mathcal{M}(\alpha_2), \mathcal{M}(\alpha_2), ..., \mathcal{M}(\alpha_i), ..., \mathcal{M}(\alpha_N)\} \\
B &= span\{\mathcal{Q}(\beta_1), \mathcal{Q}(\beta_2), \mathcal{Q}(\beta_2), ..., \mathcal{Q}(\beta_i), ..., \mathcal{Q}(\beta_N)\}
\end{aligned}
\tag{6}
$$

The $\mathcal{M}(\alpha_i), \mathcal{Q}(\beta_j) \in R^D$ and $\|\mathcal{M}(\alpha_i)\| = 1$ , $\|\mathcal{Q}(\beta_j)\| = 1$, $i, j = 1, 2, 3, ..., N$. $< \mathcal{M}(\alpha_i), \mathcal{Q}(\beta_j) >$ is the cosine between $\mathcal{M}(\alpha_i)$ and $\mathcal{Q}(\beta_j)$ , $j = 1, 2, 3, ..., N$.

**Suppose**: $\exists (\alpha_a, \beta_a), (\alpha_b, \beta_b) \in \{(\alpha_i, \beta_j), i, j = 1, 2, 3, ..., N\}$ and $a \neq b$ makes:

$$
\begin{aligned}
< \mathcal{M}(\alpha_a), \mathcal{Q}(\beta_a) > &= \underset{\beta_i = \beta_a}{\arg \max} < \mathcal{M}(\alpha_a), \mathcal{Q}(\beta_i) > \\
< \mathcal{M}(\alpha_b), \mathcal{Q}(\beta_b) > &< \underset{\beta_j \neq \beta_b}{\arg \max} < \mathcal{M}(\alpha_b), \mathcal{Q}(\beta_j) >
\end{aligned}
\tag{7}
$$

### 7.1.1 INTER-MODAL DEVIATION LEADS TO COGNITIVE DISORDER

**Prove**: There is a rotation matrix pair $(\mathcal{A},\mathcal{B})$ that not only keeps the A and B topology unbiased and makes the

$$
\begin{aligned}
< \mathcal{M}'(\alpha_a), \mathcal{Q}'(\beta_a) > &< \underset{\beta_i \neq \beta_a}{\arg \max} < \mathcal{M}'(\alpha_a), \mathcal{Q}'(\beta_i) > \\
< \mathcal{M}'(\alpha_b), \mathcal{Q}'(\beta_b) > &= \underset{\beta_j = \beta_b}{\arg \max} < \mathcal{M}'(\alpha_b), \mathcal{Q}'(\beta_j) >
\end{aligned}
\tag{8}
$$

where the $\mathcal{M}' = \mathcal{A}(\mathcal{M})$ and $\mathcal{Q}' = \mathcal{B}(\mathcal{Q})$, $\mathcal{A} \neq \mathcal{B}$. And the space A and B can be written as $A'$ and $B'$:

$$
\begin{aligned}
A' &= \mathcal{A}(A) = span\{\mathcal{M}'(\alpha_1), \mathcal{M}'(\alpha_2), \mathcal{M}'(\alpha_2), ..., \mathcal{M}'(\alpha_i), ..., \mathcal{M}'(\alpha_N)\} \\
B' &= \mathcal{B}(B) = span\{\mathcal{Q}'(\beta_1), \mathcal{Q}'(\beta_2), \mathcal{Q}'(\beta_2), ..., \mathcal{Q}'(\beta_i), ..., \mathcal{Q}'(\beta_N)\}
\end{aligned}
\tag{9}
$$

**Solution**: the Equ.7 can be written as:

$$
\begin{aligned}
< \mathcal{M}(\alpha_a), \mathcal{Q}(\beta_a) > - < \mathcal{M}(\alpha_a), \mathcal{Q}(\beta_i) >> 0, \forall \beta_i \in \beta, i \neq a \\
< \mathcal{M}(\alpha_b), \mathcal{Q}(\beta_b) > - < \mathcal{M}(\alpha_b), \mathcal{Q}(\beta_j) >< 0, \exists \beta_j \in \beta, j \neq b
\end{aligned}
\tag{10}
$$

hence:

$$
\begin{aligned}
\mathcal{M}(\alpha_a)^T \mathcal{Q}(\beta_a) - \mathcal{M}(\alpha_a)^T \mathcal{Q}(\beta_i) > 0, \forall \beta_i \in \beta, i \neq a \\
\mathcal{M}(\alpha_b)^T \mathcal{Q}(\beta_j) - \mathcal{M}(\alpha_b)^T \mathcal{Q}(\beta_b) > 0, \exists \beta_j \in \beta, j \neq b
\end{aligned}
\tag{11}
$$

because the rotation matrix pair $(\mathcal{A},\mathcal{B})$ can be seen as a rotation matrix $\mathcal{R}(\theta^D)$, where the $\theta^D$ is a rotation angle between AB and $A'B'$. Hence, when applying this rotation matrix $\mathcal{R}(\theta^D)$, the Equ.11 can be written as:

$$\mathcal{M}(\alpha_a)^T \mathcal{R}(\theta^D) \mathcal{Q}(\beta_a) - \mathcal{M}(\alpha_a)^T \mathcal{R}(\theta^D) \mathcal{Q}(\beta_i) < 0, \exists \beta_i \in \beta, i \neq a$$
$$\mathcal{M}(\alpha_b)^T \mathcal{R}(\theta^D) \mathcal{Q}(\beta_j) - \mathcal{M}(\alpha_b)^T \mathcal{R}(\theta^D) \mathcal{Q}(\beta_b) < 0, \forall \beta_j \in \beta, j \neq b \tag{12}$$

Because the rotation matrix satisfies that the inner product of itself is 1. So, Equ 12 can be written as:

$$\mathcal{M}(\alpha_a)^T \mathcal{R}(\theta^D)(\mathcal{Q}(\beta_a) - \mathcal{Q}(\beta_i)) < 0, \exists \beta_i \in \beta, i \neq a, \mathcal{R}(\theta^D)^T \mathcal{R}(\theta^D) = \mathcal{I} \tag{13}$$

$$\mathcal{M}(\alpha_b)^T \mathcal{R}(\theta^D)(\mathcal{Q}(\beta_j) - \mathcal{Q}(\beta_b)) < 0, \forall \beta_j \in \beta, j \neq b, \mathcal{R}(\theta^D)^T \mathcal{R}(\theta^D) = \mathcal{I} \tag{14}$$

For example, when $\mathcal{R}(\theta^D) = -\mathcal{I}$, then $\mathcal{R}(\theta^D)^T \mathcal{M}(\alpha_a) = -\mathcal{M}(\alpha_a)$ the equ 15 and 16 will hold. So, rotation matrices $(\mathcal{A}, \mathcal{B})$ that makes Equ.8 true exists.

### 7.1.2 Intra-modal Rotation leads To Cognitive Disorder

Since intra-modal rotation just requires the length of representation vectors after rotation is 1 and does not require that the intra-modal representation space is invariant, it is a more general case of inter-modal deviation. This means that all rotation matrixes that satisfy 7.1.1 can also satisfy Intra-modal Rotation. Different from intra-modal deviation, the inner product of the mapping matrix $\mathcal{P}$ does not require to be 1. So, we rewrite the Equ 15 and 16 to:

$$\mathcal{M}(\alpha_a)^T (\mathcal{Q}(\beta_a) - \mathcal{Q}(\beta_i))\mathcal{P} < 0, \exists \beta_i \in \beta, i \neq a, \tag{15}$$

$$\mathcal{M}(\alpha_b)^T (\mathcal{Q}(\beta_j) - \mathcal{Q}(\beta_b))\mathcal{P} < 0, \forall \beta_j \in \beta, j \neq b, \tag{16}$$

Any mapping matrix $\mathcal{P}$ that rotation the direction of $(\mathcal{Q}(\beta_j) - \mathcal{Q}(\beta_b))$ by more than 90 degrees.

### 7.2 Detailed SAM and RAM Distribution of Language Encoders

The topological structure of the language representation space does not change significantly with the continual CLIP training. But the whole language representation space, like the vision representation space, has a large rotation around the center of the high-dimensional sphere during the continual training. The angle change distribution table 7(a) and rotation angle distribution table 7(b) are shown below.

| $\theta_{SAM} \in$ | $[0°, 5°]$ | $(5°, 10°]$ | $(10°, 15°]$ | $(15°, 20°]$ | $(20°, 180°]$ |
|---|---|---|---|---|---|
| $SAM_{0-1}$ | 64.43% | 28.49% | 6.23% | 0.78% | 0.07% |
| $SAM_{1-2}$ | 71.54% | 24.89% | 3.35% | 0.22% | 0.01% |
| $SAM_{2-3}$ | 71.36% | 25.01% | 3.40% | 0.22% | 0.01% |
| $SAM_{3-4}$ | 67.30% | 27.27% | 4.93% | 0.48% | 0.03% |
| $SAM_{4-5}$ | 58.84% | 30.70% | 8.77% | 1.50% | 0.20% |
| $SAM_{0-5}$ | 55.39% | 31.60% | 10.52% | 2.15% | 0.33% |

(a) The angle difference distribution in text representation space.

| $\theta_{RAM} \in$ | $[0°, 15°]$ | $(15°, 20°]$ | $(20°, 25°]$ | $(25°, 30°]$ | $(30°, 180°]$ |
|---|---|---|---|---|---|
| $RAM_{(0,1)}$ | 0.00% | 1.94% | 28.38% | 45.88% | 23.80% |
| $RAM_{(1,2)}$ | 0.02% | 8.90% | 47.76% | 34.94% | 8.38% |
| $RAM_{(2,3)}$ | 0.04% | 1.14% | 49.86% | 31.18% | 7.52% |
| $RAM_{(3,4)}$ | 0.02% | 2.84% | 33.70% | 43.76% | 19.68% |
| $RAM_{(4,5)}$ | 0.00% | 0.04% | 3.28% | 27.66% | 69.02% |
| $RAM_{(0,5)}$ | 0.00% | 0.00% | 0.00% | 1.12% | 98.88% |

(b) The rotation angle distribution in text representation space.

Figure 7: Detailed SAM and RAM Distribution of Language Encoders.

By observing the table in Table 7, we can find that more than 88% of the angle change between any two language representation vectors in the language representation space are between 0 and 10 degrees in the process of continual CLIP training, while only 20% are above 10 degrees. Moreover,

less than 0.2% of the angle changes is above 20 degrees. Those angle change between 15-20 degrees also only account for about 1.5% of all images pairs. Similar to the visual representation space, the direction of the same sample in the language representation space of different training phases also has changed greatly. However, unlike most of the rotations in the vision representation space, which are distributed over 30 degrees, in the language space, the rotations in the representation space are mostly distributed between 20 and 30 degrees. Because of this difference, the alignment of the CLIP for different modalities of the same sample deviates during the continual training.

### 7.3 The Relationship Between Contrastive Matrix, Intra-modal Rotation, Inter-modal Deviation and Mod-X

From a detailed point of view, the element $M_{i,j}$ in the $i,j$ position of the contrastive matrix $M$ is the similarity score of the $i$'th sample vision embedding and the $j$'th sample text embedding. Since the length of the representation vector is **1**, the similarity score $M_{i,j}$ also refers to the angle between the $i$'th sample vision embedding and the $j$'th sample text embedding. Greater similarity means a smaller angle. Therefore, the value of the diagonal elements in the contrast matrix $M$ represents the angle between different modals of the same sample. The value of the off-diagonal elements represents the angle between the different modals of different samples in the CLIP's representation space. Through our exploration (in section 3), the Intra-modal Rotation and the Inter-modal Deviation affect these angles or similarity scores. From an overall perspective, **the similarity distribution of the contrastive matrix $M$ is equivalent to the structure of the representation space of the model.** Our Mod-X framework attempts to distill the similarity distribution of off-diagonal elements identical to distilling the model's representation space structure, which reduces the influence of intra-modal rotation and inter-modal deviation during continual CLIP training.

To better illustrate the relationship between the model's representation space and the model's similarity performance, we add a more direct statistical analysis, inter-modal angle variation distribution. Based on the settings in section 3, in the training phase $t$, we compare the change of angle distribution between modalities for the training samples retrieved correctly in the training phase $t-1$. A schematic diagram of inter-modal angle variation $\theta_{ImAV}$ is shown in Figure 8(a), where the sample $a$ refers to the training sample that can be retrieved correctly by model $\text{CLIP}_{t-1}$ in training phase $t-1$. The $V$ is the vision representation and $L$ is the language representation. Inter-modal angle variation distribution table can be seen in Figure 8(b).

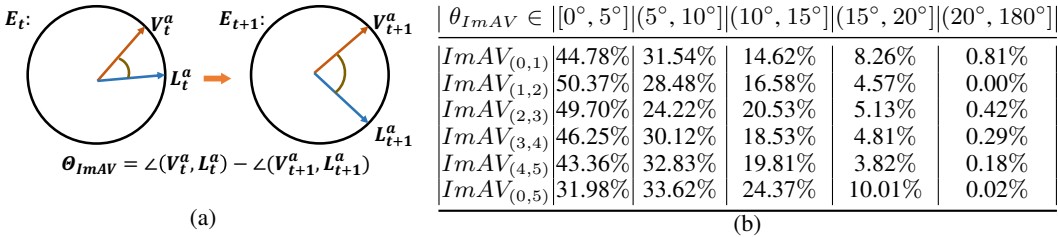

| $\theta_{ImAV} \in$ | $[0°, 5°]$ | $(5°, 10°]$ | $(10°, 15°]$ | $(15°, 20°]$ | $(20°, 180°]$ |
|---|---|---|---|---|---|
| $ImAV_{(0,1)}$ | 44.78% | 31.54% | 14.62% | 8.26% | 0.81% |
| $ImAV_{(1,2)}$ | 50.37% | 28.48% | 16.58% | 4.57% | 0.00% |
| $ImAV_{(2,3)}$ | 49.70% | 24.22% | 20.53% | 5.13% | 0.42% |
| $ImAV_{(3,4)}$ | 46.25% | 30.12% | 18.53% | 4.81% | 0.29% |
| $ImAV_{(4,5)}$ | 43.36% | 32.83% | 19.81% | 3.82% | 0.18% |
| $ImAV_{(0,5)}$ | 31.98% | 33.62% | 24.37% | 10.01% | 0.02% |

(a)         (b)

Figure 8: The sub-figure on the left shows a schematic diagram of computing $\theta_{ImAV}$. The table on the right shows the distribution of the change of the included angle of the vision and language representation of the samples, which were correctly retrieved in the previous training phase.

As shown in Figure 8(b), during the continual training, the samples that were correctly retrieved in the past have apparent changes in the angle between the modalities as the training phases go up. Only less than 50% of the samples change within 5 degrees in the continual training, and about 30% of the samples have a change of 5-10 degrees. However, more than 20% of the samples change their included angle by more than 10 degrees during the training process. This shows that the inter-modal spatial alignment (similarity performance) of the $\text{CLIP}_{ct}$ is affected by Intra-modal Rotation and Inter-modal Deviation.

To illustrate our Mod-X framework indeed alleviates the distribution shift in representation space between sample modalities during continual training, we show the inter-modal angle variation distribution of the $\text{CLIP}_{Mod-X}$ in Experiment A in Table 3.

Comparing the Figure 8(b) and Table 3, it can be found that the $\text{CLIP}_{Mod-X}$ well maintains the inter-modal spatial alignment of the correctly retrieved samples during the continual CLIP training.

| $\theta_{ImAM} \in$ | $[0°, 5°]$ | $(5°, 10°]$ | $(10°, 15°]$ | $(15°, 20°]$ | $(20°, 180°]$ |
|---|---|---|---|---|---|
| $ImAM_{(0,1)}$ | 88.66% | 7.81% | 2.56% | 0.97% | 0.00% |
| $ImAM_{(1,2)}$ | 91.79% | 4.01% | 3.20% | 0.00% | 0.00% |
| $ImAM_{(2,3)}$ | 90.70% | 9.02% | 0.24% | 0.04% | 0.01% |
| $ImAM_{(3,4)}$ | 92.13% | 6.20% | 1.61% | 0.06% | 0.00% |
| $ImAM_{(4,5)}$ | 91.91% | 7.71% | 0.38% | 0.00% | 0.00% |
| $ImAM_{(0,5)}$ | 87.81% | 10.87% | 1.12% | 0.20% | 0.00% |

Table 3: The table shows the distribution of the change of the included angle of the vision and language representation of the $CLIP_{Mod-X}$ in Experiment A.

On average, 90% of the correctly retrieved samples have an angle change of less than 5 degrees in continual training, and the samples with an angle change of more than 15 degrees account for less than 1% of all samples. All of this shows that the Mod-X framework mitigates the cognitive disorder during continual CLIP training by preserving the inter-modal spatial alignment of the samples retrieved correctly in the past during the continual training.

### 7.3.1 VALIDATION OF INTER-MODAL DEVIATION ON DIFFERENT TRAINING DATASETS

In section 3, we discuss the representational space variation of $CLIP_{ct}$ under the open-world dataset COCOLin et al. (2014) and Flickr30KYoung et al. (2014). In there, following the explore settings of the section 3.2.2, we compare the rotation distribution of the representation space of the vision and language extractors during the continual CLIP training on a specific e-commerce text to image dataset ECommerce-T2I Yang et al. (2021). The ECommerce-T2I contains 90k training images and 5k testing images set. Each image corresponds to a description, and the descriptions of training set and test set do not overlap. We splitting ECommerce-T2I training dataset randomly and evenly into five sub-datasets and sequentially training them to update pre-trained model $CLIP_0$ with ViT-32/B OpenAI continually. By evaluating the rotation distribution of the modal's representation space at various training phases on the COCO(5K) testset, we drawn the rotation distribution comparison diagram in Figure 9.

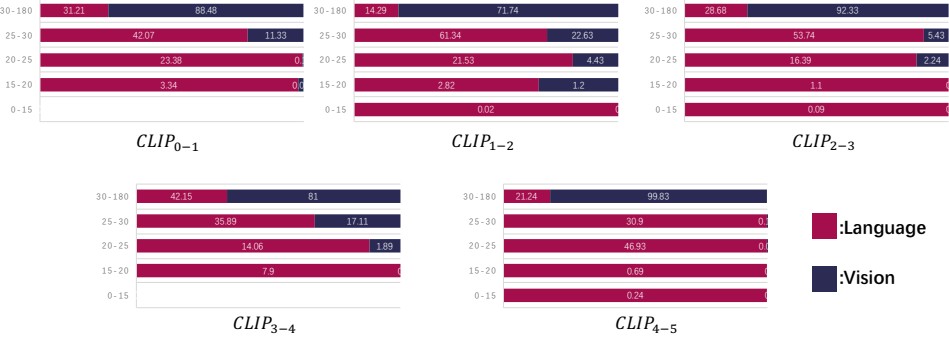

Figure 9: The comparison of the rotation distributions of the representation spaces of the vision and language extractors during continual CLIP training on ECommerce-T2I. $CLIP_{i-j}$ refers to the CLIP's continual training from training phase $i$ to $j$.

From Figure 9, we can find that when the CLIP is trained on a specific data domain, the rotation of visual representation space becomes more severe, among which more than 70% of the samples have more than 30 degrees of rotation in the visual space, which is higher than that of the open-world dataset. Although the rotation of more than 30 degrees in the language space has also seen a large proportional increase than the open-world dataset, it is still significantly out of sync with the rotation in the visual space. Most samples are rotated within 30 degrees in language space. Through this validation, we show that inter-modal deviation (rotational asynchrony) of the representation space of different modal encoders persists during the continual CLIP training on a specific data domain.

## 7.4 DETAILED EXPERIMENT SETTING

In exploration experiments 3 and main experiments 5, we use RN50 He et al. (2016) as the vision encoder and language encoder is a transformer-based architecture which follows modification proposed in CLIP OpenAI. The input images are resized to 224 × 224 and the input texts are tokenized by WordPiece with a maximum length of 77. We utilize AdamW Loshchilov & Hutter (2017) optimizer and a cosine annealing learning rate schedule with warmup which is consistent with OpenAI. All of the experiments are conducted on 8 NVIDIA V100 GPUS. In exploration Experiment and Experiment A, we use the hyper-parameters as be shown in table 3(a). Since the size of training data in our exploration experiment and experiment A is relatively small compared to large-scale pre-training, we set a smaller batch size. And other hyper-parameters is consistent with CLIP OpenAI.

(a)

| Hyperparameter | Value |
|---|---|
| Batch size | 280 |
| Vocabulary size | 49408 |
| Training epochs | 35 |
| Initial temperature $\tau$ | 0.07 |
| $\alpha$ | 20 |
| Weight decay | 0.2 |
| Warm-up iterations (%) | 20 |
| Learning rate | $5e^{-4}$ |
| Adam $\beta_1$ | 0.9 |
| Adam $\beta_2$ | 0.99 |
| Adam $\epsilon$ | $1e^{-8}$ |

(b)

| Hyperparameter | Value |
|---|---|
| Batch size | 800 |
| Vocabulary size | 49408 |
| Training epochs | 35 |
| Initial temperature $\tau$ | 0.07 |
| $\alpha$ | 20 |
| Weight decay | 0.2 |
| Warm-up iterations (%) | 20 |
| Learning rate | $1e^{-3}$ |
| Adam $\beta_1$ | 0.9 |
| Adam $\beta_2$ | 0.99 |
| Adam $\epsilon$ | $1e^{-8}$ |

Table 4: Table (a) is the hyperparameter in exploration experiment (Section 3) and Experiment A (Section 5.3). Table (b) is the hyperparameter in Experiment B (Section 5.4).

In Experiment B, due to the size of training data reaches to 1 million, we increase the batch size to 800 and increase the initial learning rate to $1e^{-3}$. And other hyper-parameters is consistent with Experiment A. The detailed hyper-parameters in table 3(b).

### 7.4.1 THE SENSITIVITY OF HYPER-PARAMETER $\alpha$

In this section, we discuss the effect of different $\alpha$ on the final performance of the $\text{CLIP}_{Mod-X}$ based on the settings of Experiment A 5.3. Table 5 presents the final retrieval results of the $\text{CLIP}_{Mod-X}$ model with $\alpha = 10, 15, 20, 25, 30$.

| Pretraining Dataset | Model | Image-Text Retrieval(%) | | | | | | Text-Image Retrieval(%) | | | | | |
|---|---|---|---|---|---|---|---|---|---|---|---|---|---|
| | | Flickr30K(1K) | | | COCO(5K) | | | Flickr30K(1K) | | | COCO(5K) | | |
| | | R@1 | R@5 | R@10 | R@1 | R@5 | R@10 | R@1 | R@5 | R@10 | R@1 | R@5 | R@10 |
| COCO | $\text{CLIP}_0$ | 16.9 | 37.0 | 46.2 | 14.7 | 34.2 | 47.0 | 12.0 | 30.0 | 41.0 | 10.6 | 29.6 | 41.0 |
| | $\text{CLIP}_{ct}$ | 20.6 | 42.8 | 56.4 | 6.2 | 17.8 | 26.1 | 16.1 | 38.5 | 50.4 | 4.7 | 14.3 | 21.8 |
| | $\alpha = 10$ | 25.7 | 50.4 | 60.3 | 11.6 | 28.4 | 30.9 | 17.3 | 40.2 | 54.6 | 7.9 | 20.9 | 34.7 |
| | $\alpha = 15$ | 28.1 | 54.3 | 66.7 | 14.0 | 32.8 | 45.4 | 20.7 | 45.8 | 58.0 | 9.7 | 26.0 | 36.4 |
| | $\alpha = 20$ | **27.9** | **53.4** | **64.4** | **14.5** | **34.0** | **46.1** | **20.2** | **45.0** | **57.2** | **10.1** | **26.4** | **37.4** |
| | $\alpha = 25$ | 26.6 | 52.8 | 62.3 | 14.5 | 34.8 | 46.7 | 20.2 | 44.7 | 57.0 | 10.0 | 27.7 | 38.1 |
| | $\alpha = 30$ | 25.5 | 51.7 | 61.8 | 14.7 | 35.0 | 47.1 | 18.4 | 42.8 | 55.5 | 10.2 | 27.0 | 38.3 |
| COCO+F30K | $\text{CLIP}_{jt}$ | 30.1 | 55.9 | 60.1 | 16.1 | 38.1 | 51.9 | 22.5 | 48.5 | 59.6 | 11.7 | 30.9 | 42.7 |

Table 5: The final multimodal retrieval performance of different $\alpha$ on continual $\text{CLIP}_{Mod-X}$ training in the Experiment A.

From the table, we can find that although different $\alpha$ affects the performance of the $\text{CLIP}_{Mod-X}$, **different $\alpha$ does not significantly affect the effectiveness of the Mod-X framework.** The performance of $\text{CLIP}_{Mod-X}$ is better than $\text{CLIP}_{ct}$ under different $\alpha$. As $\alpha$ increases, the $\text{CLIP}_{Mod-X}$

better maintains its retrieval ability on past COCO samples. The Image-Text R@1 and Text-Image R@1 on COCO(5K) remain around 14.5% and 10.0%. However, an excessively large $\alpha$ also limits the model's ability to fit new datasets. With the value of $\alpha$ increased from 20 to 30, the Image-Text R@1 and Text-Image R@1 of the CLIP$_{Mod-X}$ on the Flickr30k(1K) drops from 27.9% and 20.2% to 25.2% and 18.4%.

## 7.5 THE DETAILED PERFORMANCE OF THE MODELS AT EACH TRAINING PHASE IN EXPERIMENT A

In figure 10 , we show the effect of our framework Mod-X (CLIP$_{Mod-X}$) in each training phase and compare the performance with continual CLIP training strategies in Experiment A 5.3. From the

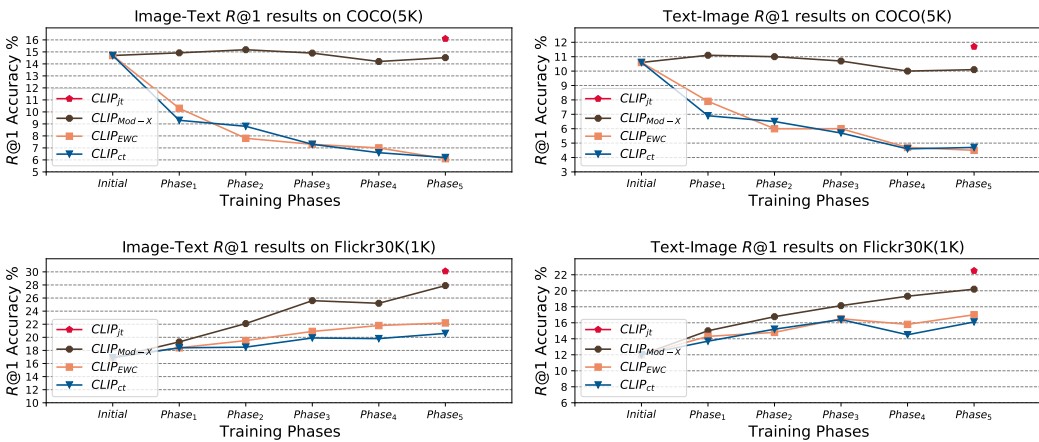

Figure 10: The performance of different training strategies in each training phase in Experiment A.

results of multimodal retrieval at each training phase, we can find that our framework, Mod-X, still has a good performance on the past pre-training dataset COCO(5K) during the continual training on the Flickr30K dataset. At each training phase, the $R@1$ results of CLIP$_{Mod-X}$ on COCO(5K) did not show a significant drop, and the gap with the initial accuracy (CLIP$_0$) remained at $\pm 1\%$. In addition to this, by comparing the retrieval performance of the CLIP$_{ct}$ and CLIP$_{Mod-X}$ on the current training data domain (Flickr30K), it can be found that the CLIP$_{Mod-X}$ is also significantly better than CLIP$_{ct}$ in continual fitting the current data domain. The low performance of traditional regularization methods EWC also shows that continual multimodal training is more complex than single-modal supervised training.

## 7.6 THE DETAILED PERFORMANCE OF THE MODELS AT EACH TRAINING PHASE IN EXPERIMENT B

In figure 11, we show the effect of our framework Mod-X (CLIP$_{Mod-X}$) in each training phase in large-scale continual training settings (the Experiment B 5.4).

Comparing the $R@1$ results of the three continual training strategies at each training phase, we can clearly see that our Mod-X framework performance in each training phase is stabilized. As the training phase continues to rise, the performance of the Mod-X improves relative to the initial pre-training results (initial). The Image-Text and Text-Image $R@1$ results on COCO(5K) had risen by 1.7% and 0.5% points, respectively. And the gap of Image-Text $R@1$ results on the Flickr30K(1K) between CLIP$_{jt}$ and CLIP$_{Mod-X}$ narrowed from 6.3% to 2.0%. The performance of Text-Image $R@1$ on Flickr30K(1K) also improved to 25.98% from 23.44%. Conversely, the ability of cognition alignment in CLIP$_{ct}$ and CLIP$_{EWC}$ are lost in the continual large-scale training. This results in verifying that our framework Mox-X can iterate under continual large-scale training and can maintain or even improve the model's cognitive ability of past entity concepts.

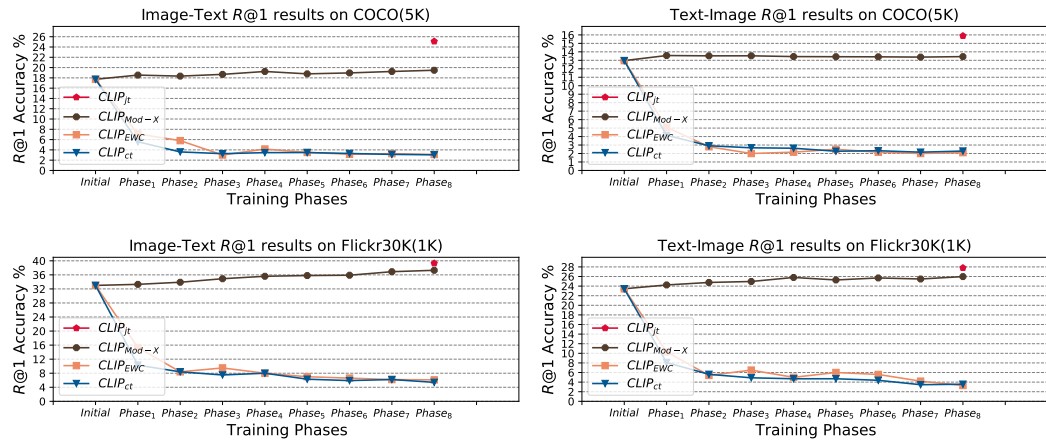

Figure 11: The performance of different training strategies in each training phase in Experiment B.

## 7.7 THE PERFORMANCE OF THE MOD-X WHEN TRAINING IN CC12M DATASET

In this section, we show the performance of different continual training strategies in CC12M Chang-pinyo et al. (2021) training dataset. The CC12M training dataset collects about 12M images and their raw descriptions harvested from the alt-text HTML attribute associated with the webscraped images, therefore representing a wider variety of content styles. Due to unavailable URLs, we utilize about 10M examples from this dataset. Firstly, we randomly and evenly split the CC12M dataset into 10 sub-datasets, each containing 1M image-text pairs. Then, we continuously train a CLIP based on these sub-datasets from scratch without any pre-training. **The purpose of this experiment is to demonstrate that the Mod-X framework still excels in large-scale continual pre-training.** In table 6, we show the final retrieval performance of different continual training strategies in COCO(5K) and Flickr30K(1K) test sets. The $CLIP_{ct}$ means continual training without any other operations. The $CLIP_{Mod-X}$ means continual training using our Mod-X framework. And the $CLIP_{jt}$ refers to training CLIP model using the joint dataset CC12M.

| Model | Image-Text Retrieval(%) | | | | | | Text-Image Retrieval(%) | | | | | |
| | Flickr30k(1K) | | | COCO(5K) | | | Flickr30k(1K) | | | COCO(5K) | | |
| | R@1 | R@5 | R@10 | R@1 | R@5 | R@10 | R@1 | R@5 | R@10 | R@1 | R@5 | R@10 |
|---|---|---|---|---|---|---|---|---|---|---|---|---|
| $CLIP_{ct}$ | 35.50 | 64.80 | 76.10 | 17.38 | 39.24 | 51.68 | 24.54 | 49.96 | 61.44 | 12.10 | 29.60 | 40.26 |
| $CLIP_{Mod-X}$ | **40.40** | **67.90** | **77.40** | **22.06** | **46.12** | **58.14** | **27.74** | **53.88** | **64.66** | **14.22** | **33.68** | **45.02** |
| $CLIP_{jt}$ | 58.00 | 83.90 | 90.40 | 34.38 | 60.30 | 71.50 | 43.02 | 72.34 | 80.92 | 22.63 | 46.44 | 58.35 |

Table 6: The final multimoal retrieval performance of the $CLIP_{ct}$, $CLIP_{Mod-X}$ and $CLIP_{jt}$ on COCO(5K) and Flickr30K(1K).

Comparing the final performance of the three training strategies, Mod-X framework ($CLIP_{Mod-X}$) still outperforms $CLIP_{ct}$ in the large-scale pre-training. After continual pre-training, the $CLIP_{Mod-X}$ obtain 40.40% Image-Text R@1 result and 27.74% Text-Image R@1 result on Flickr30K(1K) test set, which surpasses the 35.50% and 24.54% of $CLIP_{ct}$. The results on COCO(5K) are similar to those on Flickr30K(1K). The Image-Text R@1 result of $CLIP_{Mod-X}$ on COCO(5K) is 4.68% points higher than $CLIP_{ct}$ and the Text-Image R@1 result of $CLIP_{Mod-X}$ on COCO(5K) exceeds $CLIP_{ct}$ 2.12% points. The detailed R@1 performance of three training strategies at each training phase can be seen in Figure 12.

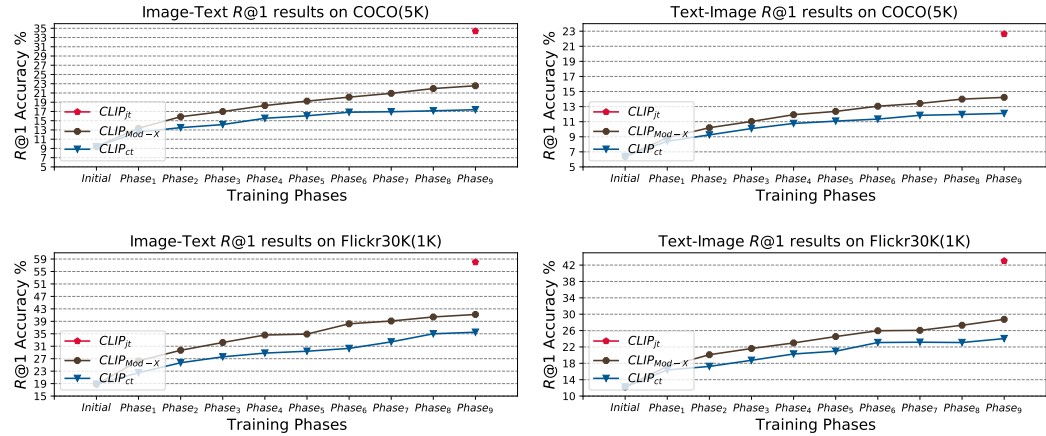

Figure 12: The retrieval performance of different training strategies in each training phase on COCO(5K) and Flickr30K(1K).

## 7.8 THE PERFORMANCE OF MOD-X WHEN FINE-TUNING THE OPENAI'S CLIP

In this section, we simulate continual CLIP training based on OpenAI's pre-trained model $\text{CLIP}_{vit32}$ with ViT-32/B vision encoder OpenAI.

### 7.8.1 THE PERFORMANCE OF MOD-X WHEN FINE-TUNING THE OPENAI'S CLIP ON COCO AND FLICKR30K DATASET

We set the $\text{CLIP}_{vit32}$ as the initial model in the continuous training process and divide the joint-dataset (COCO and Flickr30K) into five sub-datasets uniformly and randomly to simulate streaming data. Because the pre-training datasets of $\text{CLIP}_{vit32}$ are not available, we train $\text{CLIP}_{vit32}$ on the joint-dataset to get the model $\text{CLIP}_{ft}$ as an upper bound for the performance of continual training. We apply our framework Mod-X in this setting and compare the final multimodal retrieval results with $\text{CLIP}_{ct}$, which is just continual training without any other operations, in Table 7.

| Model | Image-Text Retrieval(%) | | | | | | Text-Image Retrieval(%) | | | | | |
| | Flickr30k(1K) | | | COCO(5K) | | | Flickr30k(1K) | | | COCO(5K) | | |
| | R@1 | R@5 | R@10 | R@1 | R@5 | R@10 | R@1 | R@5 | R@10 | R@1 | R@5 | R@10 |
|---|---|---|---|---|---|---|---|---|---|---|---|---|
| $\text{CLIP}_{vit32}$ | 77.7 | 94.5 | 98.3 | 50.1 | 74.6 | 83.0 | 58.9 | 83.5 | 90.1 | 30.2 | 55.6 | 66.7 |
| $\text{CLIP}_{ct}$ | 85.6 | 97.3 | 98.8 | 59.7 | 83.2 | 90.2 | 71.2 | 91.5 | 94.9 | 43.5 | 70.9 | 80.6 |
| $\text{CLIP}_{Mod-X}$ | **86.9** | **97.7** | **99.3** | **62.1** | **85.6** | **91.7** | **73.4** | **92.9** | **96.2** | **46.2** | **73.5** | **82.6** |
| $\text{CLIP}_{ft}$ | 86.3 | 97.2 | 99.1 | 63.6 | 86.4 | 92.3 | 72.7 | 92.6 | 96.3 | 46.3 | 73.1 | 82.3 |

Table 7: The final multimoal retrieval performance of the $\text{CLIP}_{ct}$, $\text{CLIP}_{Mod-X}$ and $\text{CLIP}_{ft}$ based on OpenAI's $\text{CLIP}_{vit32}$ with VIT-32/B vision encoder.

The performance of our framework Mod-X is still better than $\text{CLIP}_{ct}$ on all of the evaluation settings. Comparing the $R@1$ results on the test set Flickr30K(1K), we can find that $\text{CLIP}_{Mod-X}$ not only surpasses the initial results ($\text{CLIP}_{vit32}$) but also 1.3% points and 2.2% points higher than $\text{CLIP}_{ct}$. The results on COCO(5K) also illustrate that our framework not only resists the cognitive disorder of the model but also fits the new data domain better than $\text{CLIP}_{ct}$. The $R@1$ results of $\text{CLIP}_{Mod-X}$ on COCO(5K) surpasses the $\text{CLIP}_{ct}$ by 2.4% and 2.7% points, respectively.

### 7.8.2 THE PERFORMANCE OF MOD-X WHEN FINE-TUNING THE OPENAI'S CLIP ON ECOMMERCE-T2I DATASET

To illustrate that the Mod-X framework is not only applicable to open-world datasets, in this section, we compare the performance of $\text{CLIP}_{Mod-X}$ with $\text{CLIP}_{ct}$ and $\text{CLIP}_{ft}$ for continual training on a

specific e-commerce data domain (ECommerce-T2I). The $\text{CLIP}_{ct}$ is just continual training without any other operations. The $\text{CLIP}_{ft}$ updates $\text{CLIP}_{vit32}$ using joint-dataset ECommerce-T2I. The ECommerce-T2I Yang et al. (2021) is a text-to-image e-commerce dataset that contains 90k training images and 5k testing images set (EC(5K)). Each image corresponds to a description, and the descriptions of training set and test set do not overlap. We set the $\text{CLIP}_{vit32}$ as the initial model in the continuous training process and divide the joint-dataset ECommerce-T2I into five sub-datasets uniformly and randomly to simulate streaming data. The final multimodal retrieval results have been shown in Table 8.

| Model | Image-Text Retrieval(%) | | | | | | Text-Image Retrieval(%) | | | | | |
|---|---|---|---|---|---|---|---|---|---|---|---|---|
| | Flickr30k(1K) | | COCO(5K) | | EC(5K) | | Flickr30k(1K) | | COCO(5K) | | EC(5K) | |
| | R@1 | R@5 | R@1 | R@5 | R@1 | R@5 | R@1 | R@5 | R@1 | R@5 | R@1 | R@5 |
| $\text{CLIP}_{vit32}$ | 77.7 | 94.5 | 50.1 | 74.6 | 11.3 | 27.6 | 58.9 | 83.5 | 30.2 | 55.6 | 10.1 | 25.5 |
| $\text{CLIP}_{ct}$ | 63.4 | 87.2 | 36.8 | 61.5 | 16.6 | 40.7 | 44.4 | 71.0 | 20.6 | 42.6 | 15.8 | 40.5 |
| $\text{CLIP}_{Mod-X}$ | **73.1** | **92.1** | **47.1** | **70.5** | **20.1** | **44.8** | **55.6** | **79.9** | **27.9** | **51.0** | **20.0** | **44.8** |
| $\text{CLIP}_{ft}$ | 64.5 | 88.6 | 39.8 | 64.8 | 23.5 | 50.8 | 46.9 | 73.1 | 22.2 | 44.5 | 23.5 | 50.6 |

Table 8: The final multimoal retrieval performance of the $\text{CLIP}_{ct}$, $\text{CLIP}_{Mod-X}$ and $\text{CLIP}_{ft}$ based on OpenAI's $\text{CLIP}_{vit32}$ on specific e-commerce data domain (ECommerce-T2I).

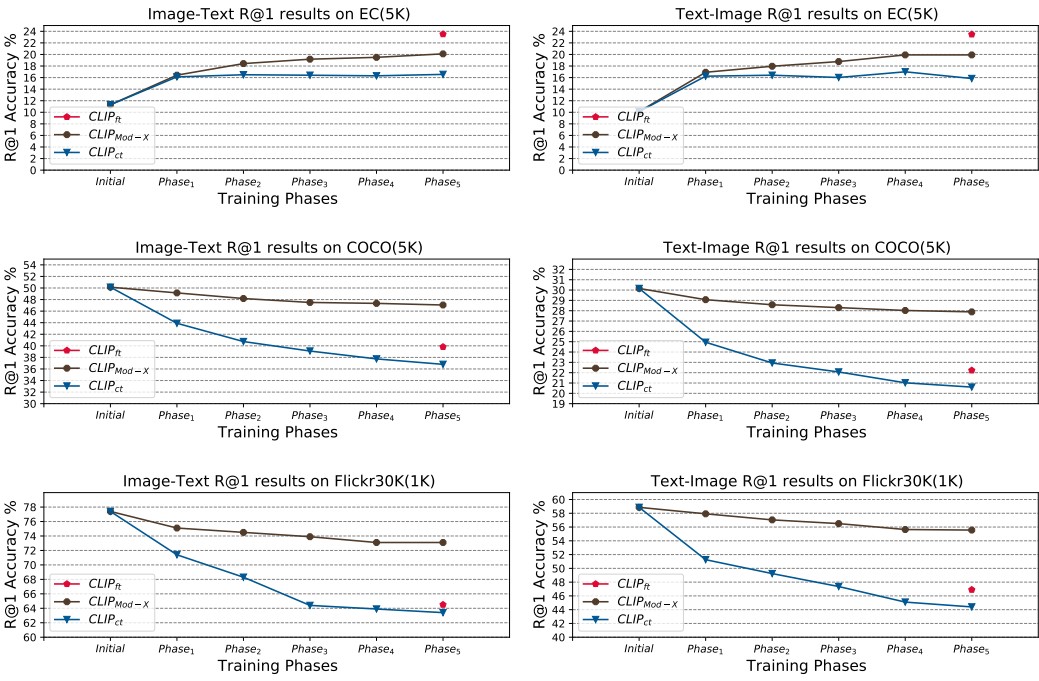

Figure 13: The $R@1$ retrieval performance of different training strategies in each training phase on EC(5K), COCO(5K) and Flickr30K(1K).

Comparing the $R@1$ performance of the $\text{CLIP}_{ft}$ and $\text{CLIP}_{vit32}$ on the EC(5K) test set, we once again show that the training of the CLIP model is affected by the training data domain. The $R@1$ results of $\text{CLIP}_{ft}$ on EC(5K) is 8.8% and 9.9% points higher than $\text{CLIP}_{vit32}$. However, the retrieval results of $\text{CLIP}_{ft}$ on COCO(5K) and Flickr30K(1K) have dropped by more than 10% points on average, which also means that fine-tuning (one phase continual training) the CLIP model will lose its ability to retrieve past samples. This is also verified by observations of the retrieval performance of $\text{CLIP}_{ct}$ performs lower than $\text{CLIP}_{ft}$. On the contrary, the $\text{CLIP}_{Mod-X}$ obtained after continual training by the Mod-X framework only has a tie drop of 3.3% points in the $R@1$ retrieval results on COCO(5K) and Flickr30K(1K). What's more, the performance of the $\text{CLIP}_{Mod-X}$ on EC(5K) outperformed $\text{CLIP}_{ct}$ by 3.5% points and 4.2% points on Image-Text R@1 and Text-Image R@1,

respectively. All of this shows that Mod-X framework not only preserves the inter-modal spatial structure of old samples during the continual training but also improves the fitting ability of the CLIP in the current training data domain. Figure 13 presents the $R@1$ retrieval performance of this three training strategies on the COCO(5K), Flickr30K and EC(5K) at each training phase. The trend of $R@1$ performance of these three training strategies during continual training on three test sets also illustrates that the Mod-X framework significantly alleviates the cognitive disorder during the continual CLIP training.

## 7.9 THE REPRESENTATION QUALITY OF VISION ENCODERS DURING CONTINUAL CLIP TRAINING

In Section 3, based on the distribution Table 2(b), we inference that the topology of the visual representation of the $CLIP_{ct}$ changes slowly during the continual CLIP training. Due to the topology of the representation space is correlated with the quality of the model's representation, so we use the linear evaluation method, commonly used in self-supervision Oord et al. (2018); He et al. (2020), to detect the quality of the model's vision encoders to verify our suppose.By fixing the vision encoder, retrain a single Linear layer, which is connected behind the vision encoder, based on the ImageNet Deng et al. (2009) training set and evaluate its top-1 accuracy on the ImageNet test set to represent the vision encoder's representation quality.

### 7.9.1 THE CHANGES IN THE REPRESENTATION QUALITY OF VISUAL ENCODERS IN EXPERIMENT A.

This section discusses the representation quality of the CLIP's visual encoders in Experiment A. As shown in Figure 14, we calculate the vision encoders' linear evaluation in different training phases.

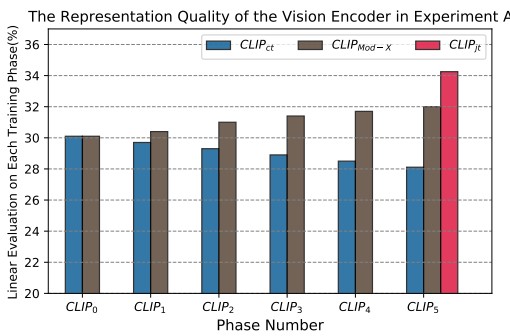

Figure 14: The changes in the representation quality of visual encoders in Experiment A.

Observing the changing trends in the linear evaluation accuracy of each training phase, we can find that the representation quality of the vision encoder in $CLIP_{cl}$ gradually decreases as the training phase increases. The top-1 accuracy in the ImageNet test set dropped from 30.1% to 28.1%, which is consistent with our conjecture 3.2.1. Compared to the decline in multimodal retrieval, the decrease in the quality of visual representations appears to be negligible. In addition, by comparing the results of $CLIP_{Mod-X}$ and $CLIP_{jt}$, we can find that our Mod-X framework can not only help the model fit new image-text samples but also improve the representation quality of the modal encoders within the CLIP. The top-1 accuracy of the vision encoder in $CLIP_{Mod-X}$ improved from 30.1% to 32.0%. All of this also illustrates that the quality of the extractor representation is not precisely positively correlated with the cognitive ability of the model.

### 7.9.2 THE CHANGES IN THE REPRESENTATION QUALITY OF VISUAL ENCODERS IN EXPERIMENT B.

This section discusses the representation quality of the CLIP's visual encoders in Experiment B. As shown in Figure 15, we calculate the vision encoders' linear evaluation in different training phases.

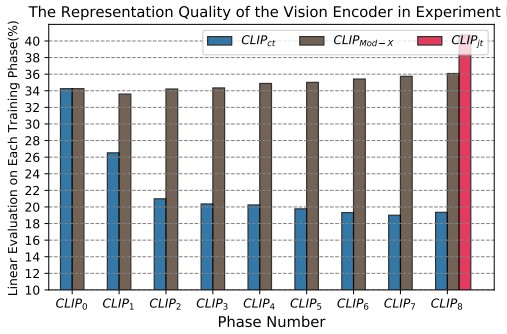

Figure 15: The changes in the representation quality of visual encoders in Experiment B.

Due to the continual large-scale training, the representation quality in $\text{CLIP}_{ct}$ has fallen off a cliff. The trend is similar to its performance on multimodal retrieval. Although our Mod-X framework maintains the quality of the modal encoders and the top-1 accuracy of vision encoders increased to 36% from 34.25%, the gap between $\text{CLIP}_{jt}$ still has 4%. We think how to shorten this gap will become a question worth considering.

