# OpenReview forum: "Continual Vision-Language Representaion Learning with Off-Diagonal Information"
_ICLR.cc/2023/Conference — Submitted to ICLR 2023_

### Official Review · Reviewer_TamL · 2022-10-23

**Confidence:** 4
**Correctness:** 3
**Technical Novelty And Significance:** 3
**Empirical Novelty And Significance:** 4
**Recommendation:** 5

**Clarity, Quality, Novelty And Reproducibility:**

Clarity.
The paper clearly introduces the re-training issue of the CLIP model. The authors investigate what correlates with performance drop on the image-text retrieval task.

Quality.
The formwork is clear and well-organized.

Novelty.
They investigate the proposed problem from the representations and record the difference across several continual learning stages, which is new. But the proposed method seems irrelevant to the findings.

Reproducibility.
Code is not provided.


**Strength And Weaknesses:**

Strength
1. The continual learning of CLIP model is challenging but very interesting.
2. Authors correlate the degrading performance with the representation variations.
3. The manuscript's structure is well-organized and easy to follow.

Weakness
1. The conclusion can not drawed from the provided observations and experiments in the paper. First, only two retrieval datasets are considered, the representation differences may vary in different datasets. Second, the correlations between the retrieval performance and representation variations are not evaluated.
2. For the proposed method, the data are different during different stages, but the training objective is modified on different training iterations. Why would the new loss alleviate the forgetting issue?
3. The experiments are not sufficient to demonstrate the claims in the introduction.

**Summary Of The Paper:**

This paper studies a continual learning problem based on the previous CLIP model. COCO and Flicker datasets are investigated to determine the representation variations when continually trained on one dataset and tested on the other.  They found that the retrieval performance on one dataset would drop significantly when continually trained. They correlate this phenomenon with representation variations. To alleviate this, they introduce a new training loss to contain the predictions between current and previous predictions.

**Summary Of The Review:**

Clarity.
 The authors investigate the image-text retrieval continual learning problem based on the CLIP model. They provide several interesting findings and modify the training objectives. But the experimental results are not sufficient to support the claims of the present paper. Therefore, I tend to reject the current version.

---

> ### Author Response · Authors · 2022-11-11
> **Response to Reviewer TamL**
>
> **Q1: The conclusion should be validated on more datasets. And the correlations between the retrieval performance and representation variations should be evaluated.**
> **A1:** Thanks for your nice suggestion. In section 3, we discuss the representational space variation of $CLIP_{ct}$ under the open-world dataset COCO and Flickr30K. In section 7.3.1, we add experiments to show that inter-modal spatial shift still exists when continual training on a specific e-commerce text-to-image dataset ECommerce-T2I. Besides this, in section 7.3, we add experiments to show the relationship between contrastive matrix, intra-modal rotation, inter-modal deviation and Mod-X. Comparing the inter-modal angle variation distribution in Figure 8(b) and Table 3, we can find that the $CLIP_{Mod-X}$ well maintains the inter-modal spatial alignment of the correctly retrieved samples during the continual CLIP training.
>
>
> **Q2: Why would the new loss alleviate the forgetting issue?**
> **A2:** Thanks for your question. From a detailed point of view, the element $M_{i,j}$ in the i,j position of the contrastive matrix M is the similarity score of the i'th sample vision embedding and the j'th sample text embedding. Since the length of the representation vector is 1, the similarity score $M_{i,j}$ also refers to the angle between the i'th sample vision embedding and the j'th sample text embedding. From an overall perspective, **the similarity distribution of the contrastive matrix M is equivalent to the structure of the representation space of the CLIP model**. Therefore, using the current training data to detect the representation space of the old model and using these representation space to guide the current model to maintain the inter-modal representation alignment of old samples retrieved correctly in the previous training phase is effective in alleviating cognitive disorder.
>
>
> **Q3 The experiments are not sufficient to demonstrate the claims in the introduction.**
> **A3:** Thanks for your suggestion. In addition to adding experiments in section 7.3.1 to show that inter-modal deviation are prevalent in continual CLIP training, we also add experiments in sections 7.7 and 7.8 to show that Mod-X framework not only has the expected performance on the large-scale CC12M dataset, but continual training on specific data domains (e-commerce dataset ECommerce-T2I) based on the pre-trained model can still obtain better results than traditional fine-tuning. If the reviewer has further questions or suggestions for the manuscript, we’d be happy to address them. Thanks!

---

> > ### Comment · Reviewer_TamL · 2022-12-04
> > **Concern on the Continual generalization**
> >
> > Thank you for your detailed response and for providing the revised version. CLIP is a very powerful pre-training model which generalizes well on multiple tasks, including text-image retrieval, image classification, QA, captioning and many others. The continual learning on the pre-training model should focus on the generalization capacity across different tasks. For instance, when continually trained on retrieval datasets, the pre-trained model would still work when fine-tuning on QA or other tasks. I think that is the most significant point for continual pre-training models.  If the experiments are only conducted on retrieval datasets, why CLIP model is necessary instead of other models that perform better on the retrieval task?

---

> > > ### Author Response · Authors · 2022-12-05
> > > **Response to Reviewer TamL**
> > >
> > > Many thanks for your prompt reply! In order to illustrate that our Mod-X framework is still superior to pure continual pre-training on different tasks, we compare the performance of the Mod-X ($CLIP_{Mod-X}$), continual learning without other operations ($CLIP_{ct}$) and baseline joint learning ($CLIP_{jt}$) on **linear probe** and **zero-shot image classification** in the following table.
> > > **Linear probe top-1 accuracy (%) on ImageNet**:
> > > Continual Training In Flickr30K:
> > > | Method | ImageNet |
> > > | ---- | ---- |
> > > |$CLIP_{jt}$|34.2|
> > > |$CLIP_{Mod-X}$|**31.9**|
> > > |$CLIP_{ct}$|28.1|.
> > >
> > > Continual Training In SBU1M:
> > > | Method | ImageNet |
> > > | ---- | ---- |
> > > |$CLIP_{jt}$|40.8|
> > > |$CLIP_{Mod-X}$|**36.2**|
> > > |$CLIP_{ct}$|19.7|
> > >
> > > Continual Training In CC12M:
> > > | Method | ImageNet |
> > > | ---- | ---- |
> > > |$CLIP_{jt}$|47.3|
> > > |$CLIP_{Mod-X}$|**41.6**|
> > > |$CLIP_{ct}$|35.1|
> > >
> > > **zero-shot image classification top-1 accuracy (%) on numerous datasets after continual training in CC12M**:
> > > | Method | Cifar10 | Caltech101 | Places365 |  ObjectNet | ImageNet | Average |
> > > | ---- | ---- | ---- | ---- | ---- | ---- | ---- |
> > > |$CLIP_{jt}$|73.1| 40.4 | 32.3 | 10.4 | 35.7 | 38.4 |
> > > |$CLIP_{Mod-X}$ | **71.2** | **35.8** | **28.7** | **8.3** | **29.8** |  **34.8** |
> > > |$CLIP_{ct}$| 64.7 | 30.2 | 23.5 | 6.2 | 23.4 | 29.6 |
> > >
> > > From the results, the continual training based on the Mod-X framework is still better than the direct continual training (ct) in its model representation ability (linear probe) and generalization performance (zero-shot classification).

---

> ### Author Response · Authors · 2022-11-16
> **Response to Reviewer TamL**
>
> Dear reviewer, we have tried to address your concerns in our earlier responses. If you have any additional questions or suggestions, we are very happy to discuss with you.

---

> ### Author Response · Authors · 2022-12-03
> **Response to Reviewer TamL**
>
> Dear reviewer, since the discussion stage is about to end, do you have any major concerns or suggestions? We are happy to discuss with you.

---

### Official Review · Reviewer_JafF · 2022-10-24

**Confidence:** 4
**Correctness:** 3
**Technical Novelty And Significance:** 3
**Empirical Novelty And Significance:** 3
**Recommendation:** 5

**Clarity, Quality, Novelty And Reproducibility:**

A bunch of “cognitive” terms aside, the paper is clear to read. The novelty is not as strong as it might seem due to the similarity of a knowledge distillation loss. I cannot judge on reproducibility.

**Strength And Weaknesses:**

The strength of the paper is that it shows an interesting phenomenon of angle shifting – the relative angle between a positive pair didn’t move as much as the absolute angle rotates. This suggests that the representation space is undergoing constant rotation during continual learning, which is an interesting insight. Also the experimental results clearly show that the proposed model is significantly better than baselines.

The main weakness of the paper is that it reads very disconnected. There is no intuition why knowledge distillation addresses angular shift, and neither are there results confirming that using knowledge distillation helps this issue by showing the angular shift before and after distillation.

The proposed model seems generic to any contrastive learning algorithm, and the whole paper could have been benchmarked on image-only continual unsupervised learning (e.g. Madaan et al. 2021), instead of image-text learning. There is nothing specific to multi-modality here, and you could treat the two vectors coming from either an image-text pair or a pair of two views of the same image.

Moreover, the paper claims novelty on “cognitive alignment”, but it is basically a knowledge distillation loss (i.e. LwF Learning without Forgetting, Li & Hoiem 2016) applied directly in the context of contrastive learning (i.e. predicting image instance labels instead class labels). Since no gradient is backproped into the old model, the KL loss is equivalent to a cross entropy loss between old and new prediction. One minor difference is that the paper manually edits the old model’s prediction if the label is wrong. This seems an ad hoc operation and there is no empirical evidence presented in the paper showing that it is necessary. In summary, the paper may run into a concern of over-claiming, as the proposed method is LwF/knowledge distillation re-purposed in the context of contrastive learning, but it is instead branded with a bunch of “cognitive” terms (“cognitive disorder”, “cognitive deviation”, “cognitive align”) with no real cognitive science intuitions or references.

Madaan et al. Representational continuity for unsupervised continual learning. ICLR 2022.
Li & Hoiem. Learning without Forgetting. ECCV 2016.

**Summary Of The Paper:**

This paper investigates continual learning of a multi-modal text-image model like CLIP. It investigates the angle shift of the learned representations throughout different learning sessions, and it also proposes a knowledge distillation type of loss to address the issue of catastrophic forgetting. Results showing that using knowledge distillation can significantly improve the CLIP continual learning. However, the paper’s model and analyses are disconnected and there is concern on whether the model is specific to CLIP or contrastive learning in general.

**Summary Of The Review:**

The introduction of a multi-modal continual learning benchmark is interesting, and the analyses are insightful, but unfortunately the model disconnects from the analyses. I suggest the authors show a direct link why the analyses inform such a model, and also evaluate whether it is possible to run on image-only continual learning benchmarks or there are things that are specific to image+text. Overall I think the paper is slightly below the bar as it is hard to find a clear hypothesis throughout.

---

> ### Author Response · Authors · 2022-11-11
> **Response to Reviewer JafF**
>
> **Q1: There is no intuition why knowledge distillation addresses angular shift, and neither are there results confirming that using knowledge distillation helps this issue by showing the angular shift before and after distillation.**
> **A1:** Thanks for pointing out this flaw. From a detailed point of view, the element $M_{i,j}$ in the i,j position of the contrastive matrix M is the similarity score of the i'th sample vision embedding and the j'th sample text embedding. Since the length of the representation vector is 1, the similarity score $M_{i,j}$ also refers to the angle between the i'th sample vision embedding and the j'th sample text embedding. Therefore, the value of the diagonal element in the contrast matrix M represents the angle between different modals of the same sample. The value of the off-diagonal element represents the angle between different modals of different samples in the model's representation space. Through our exploration (in section 3), the Intra-modal Rotation and the Inter-modal Deviation are actually affecting these angles or similarity scores. From an overall perspective, **the similarity distribution of the contrastive matrix M is equivalent to the structure of the representation space of the CLIP model**. Our Mod-X framework attempts to distill the similarity distribution of off-diagonal elements equivalent to distilling the structure of the model's representation space, which reduces the influence of intra-modal rotation and inter-modal deviation during continual CLIP training. To better illustrate this, in section 7.3, we add experiments to show the relationship between contrastive matrix, intra-modal rotation, inter-modal deviation and Mod-X. The experimental results also show in detail that our Mod-X framework indeed preserves the spatial structure of the inter-modal of the samples retrieved correctly before.
>
> **Q2: What is the difference between continual CLIP training and image-only continual self-supervised training?**
> **A2:** Good question. As described by [Hu et al.2021], [Ni et al.2021] and [Madaan et al. 2022], due to the variation of representation space in image-only continual self-supervised training is smooth, the model does not show obvious catastrophic forgetting. And the model's performance is robust. However, through our exploration in section 3, we find that **the changes of representation topology of the two modal encoders in CLIP are not obvious during the continual CLIP training. But the retrieval performance of continual CLIP training still has a significant drop.** Through thorough analysis, we illustrate another cause of model performance degradation in continual CLIP training, namely the inter-modal deviation, which are specific to continual CLIP training. Besides this, there is still a robust underlying correlation between two views of the same image. This correlation helps the image-only self-supervised model to synchronize the representation space between the two image encoders during continual training. However, the vision and language encoders of the CLIP model do not encode the same modal information. The connection between the two modal encoders only relies on contrastive loss. Therefore they are very susceptible to training data and are more prone to inter-modal deviation in continual training.
> [Hu et al.2021]: How Well Does Self-Supervised Pre-Training Perform with Streaming Data?
> [Madaan et al. 2022]: Representational continuity for unsupervised continual learning.
> [Ni et al.2021]: Self-Supervised Class Incremental Learning.
>
> **Q3: Where is the novelty in your manuscript?**
> **A3:** Thanks for your nice question. The novelty of our work is that **1)**. We discover the performance in continual CLIP training are different with image-only continual self-supervised training. Continual CLIP training leads to persistent performance degrades on multimodal retrieval. **2)**.Through our exploration in section 3, we find the reasons that lead to CLIP's cognitive disorder are intra-modal rotation and inter-modal deviation which are specific to continual CLIP training. **3)**. The novelty of Mod-X is that, based on the exploration and theoretical analysis, we correlate the elements distribution of the contrastive matrix with the inter-modal representation space structure. And using the distillation idea appropriately to effectively alleviate the spatial shifts in continual CLIP training. The final performance in section 5 and Appendix also confirms that maintaining the correct inter-modal spatial structure mitigates the degradation in the CLIP's retrieval performance during continual training. We believe our work has scientific beauty and is of great value to be a good baseline for future multi-modal continual pre-training.

---

> ### Author Response · Authors · 2022-11-16
> **Response to Reviewer JafF**
>
> Dear reviewer, we have tried to address your concerns in our earlier responses. If you have any additional questions or suggestions, we are very happy to discuss with you.

---

> ### Author Response · Authors · 2022-12-03
> **Response to Reviewer JafF**
>
> Dear reviewer, since the discussion stage is about to end, do you have any major concerns or suggestions? We are happy to discuss with you.

---

### Official Review · Reviewer_4dCz · 2022-10-26

**Confidence:** 4
**Correctness:** 2
**Technical Novelty And Significance:** 3
**Empirical Novelty And Significance:** 2
**Recommendation:** 3

**Clarity, Quality, Novelty And Reproducibility:**

The manuscript is easy to read since the idea is very simple, but the overall writing, including the descriptions for notations, is sometimes unclear.

**Strength And Weaknesses:**

**strength**
- Introduce and analyze challenging issues for continual vision-language representation learning, named intra-modal rotation and inter-modal deviation
- outperform baselines in both image-text and text-image retrieval.

**weakness**
- Missing detailed and explicit descriptions for many notations (e.g., ct, jt, the subscript (task index) i, the superscript T in Section 3.2.3., the capital letter S in figure 6, etc
- The difference between Cognitive disorder and Catastrophic forgetting (or representational forgetting, referred to in Madaan et al. 2022) needs to be clarified.
- Intra-modal rotation may happen to all non-iid task-based problems, including streaming learning and conventional continual learning, which is not new, and the following analyses are a bit straightforward in the continual learning field.
- Including Appendix 7.1 for the mathematical derivation, inter-modal deviation is not explicitly related to cognitive disorder (the performance degeneration of retrieval tasks). It tends to, but in a strict sense, the manuscript doesn't provide evidence that increasing inter-modal deviation in each sample is always harmful to "cognitive disorder".
- Only naive and old baseline (EWC) is used for validation, which is hard to evaluate how the proposed model achieves superior performance compared to recent strong methods for multi-modal continual learning for retrieval tasks.

**Summary Of The Paper:**

The paper proposes a new method based on the CLIP model for continual vision-language representation learning. The authors find that simple continual training in the CLIP model degenerates the performance on multimodal retrieval tasks, named Cognitive Disorder (CD). To tackle CD, they suggest Mod-X that selectively aligns the off-diagonal information distribution of the contrastive matrices utilizing the CLIP models from current and past timesteps.

**Summary Of The Review:**

The direction of the submission is auspicious and interesting, but explanations might be required to be more kind and include the details. Also, I recommend further comparing with competitive methods for multimodal continual learning methods for solving retrieval tasks.

---

> ### Author Response · Authors · 2022-11-11
> **Response to Reviewer 4dCz**
>
> **Q1: Missing detailed and explicit descriptions for many notations.**
> **A1:** Thanks for pointing out the confusing notation. The $CLIP_{ct}$ means continual training without other operations, which is described in the first sentence of section 3. And $CLIP_{jt}$ means training CLIP from scratch using joint datasets, which can be found in the Exploration Setup of section 3. The superscript "T" in section 3.2.3 is a transpose operation that is often used for matrix multiplication. We have added this description in section 3.2.3. The capital letter S in figure 6 means the cosine similarity score of the i'th sample vision embedding and the j'th sample text embedding. We have added them to the caption in Figure 6. The subscript (task index) i in section 3.2 means different training phases. We have added this description in section 3.2.1.
>
> **Q2: What is the difference between the intra-modal rotation in the continual CLIP training and the representation space change of conventional continual learning? And what is the difference between Cognitive disorder and Catastrophic forgetting?**
> **A2:** Thanks for your nice question. **1)**.The main difference between intra-modal rotation in continual CLIP training and representation space change in traditional single-modal continual learning is that catastrophic forgetting, which occurs in traditional supervised continual learning, is always accompanied by **drastic** changes in the model's representation topological structure [Ni et al.2021]. Dramatic changes in representation topology also represent drastic changes in the quality of the model representation. However, in section 3, empirical experiments show that the encoders' representation space in the CLIP is significantly rotated during the continual CLIP training. But the representation topology of the encoders is slightly rotated compared with the rotation of the whole representation space. This means that the intra-modal rotation in CLIP encoders during continual CLIP training does not significantly affect the representation quality of the encoders.
> **2)**.This also brings out the difference between cognitive disorder and catastrophic forgetting. As described by [Hu et al.2021] and [Madaan et al. 2022], smoother variation of representation space in the model represents less catastrophic forgetting and more robust performance. Through our exploration in section 3, we find that the changes in representation topology of the two modal encoders in CLIP are not obvious during the continual CLIP training. However, the retrieval performance of continual CLIP training still has a significant drop. Through thorough analysis, we illustrate another cause of model performance degradation in continual CLIP training, namely the inter-modal deviation.
> [Ni et al.2021]: Revisiting Catastrophic Forgetting in Class Incremental Learning.
> [Hu et al.2021]: How Well Does Self-Supervised Pre-Training Perform with Streaming Data?
> [Madaan et al. 2022]: Representational continuity for unsupervised continual learning.

---

> > ### Author Response · Authors · 2022-11-11
> > **Response to Reviewer 4dCz**
> >
> > **Q3: What is the relationship between inter-modal deviation and cognitive disorder?**
> > **A3:** Thanks for raising an important point. The intra-modal rotation and inter-modal deviation actually affect the angle between the vision and language representation of the sample in the representation space. And since the length of the representation vector in CLIP is **1**, the angle between the vision and language representation of the sample is equivalent to the similarity between the vision and language information of the sample, that is, the diagonal element in the CLIP's contrastive matrix. The degree of cognitive disorder is equivalent to the retrieval ability of the CLIP's contrastive matrix. Therefore, an increasing inter-modal deviation aggravates the variation of inter-modal angle and is also equivalent to aggravating the cognitive disorder of the CLIP. To better illustrate this, in section 7.3, we add experiments to show the relationship between contrastive matrix, intra-modal rotation, inter-modal deviation and Mod-X.
> >
> > **Q4: Why not compare the Mod-X to recent strong methods for multi-modal continual learning for retrieval tasks?**
> > **A4:** Thanks for your suggestion. However, according to our survey, there is not much work on multi-modal continual learning. And our known methods are not suitable for a simple and open training setting like ours. [Kai Wang et al.,2021] using reindex strategy, which subsequentially indexes data in the external database to prevent degradation of cross-modal retrieval performance. However, our method does not require an external database. [Tejas Srinivasan et al.,2022] produced a continual learning benchmark
> > for Vision-and-Language tasks and used common continual learning methods for multi-modal tasks. However, it does not give its own solutions for specific multi-modal tasks. [Zhihao Fan et al.,2022] propose to use of multi-modal knowledge graphs to assist multi-modal learning, which mainly focuses on the construction of knowledge graphs. The reason why we use EWC is that it does not require an external memory bank, database, or knowledge graph to help the model's update, which is consistent with our training settings. Because our framework does not require additional settings nor any augmentation and generation operations during continual training, it is very convenient to reproduce and compare, which provides a good baseline for future multi-modal continual learning. If we lose some reproducible works in multi-modal continual pre-training, please share them with us. We will reproduce and compare them with our Mod-X framework later.
> >
> > [Kai Wang et al,2021]: Continual learning in cross-modal retrieval.
> > [Tejas Srinivasan et al,2022]: CLiMB: A Continual Learning Benchmark for Vision-and-Language Tasks.
> > [Zhihao Fan et al,2022]: A Unified Continuous Learning Framework for Multi-modal Knowledge Discovery and Pre-training.

---

> ### Author Response · Authors · 2022-11-16
> **Response to Reviewer 4dCz**
>
> Dear reviewer, we have tried to address your concerns in our earlier response. If you have any further questions or suggestions, we are very happy to discuss with you.

---

> ### Author Response · Authors · 2022-12-03
> **Response to Reviewer 4dCz**
>
> Dear reviewer, since the discussion stage is about to end, do you have any major concerns or suggestions? We are happy to discuss with you.

---

### Official Review · Reviewer_7TxY · 2022-10-29

**Confidence:** 4
**Clarity, Quality, Novelty And Reproducibility:** The paper is clear, novel and the met…
**Correctness:** 4
**Technical Novelty And Significance:** 3
**Empirical Novelty And Significance:** 4
**Recommendation:** 8

**Strength And Weaknesses:**

Strengths:
1. The paper is well written and easy to follow.
2. The paper provides a thorough analysis on the causes for performance drop in vision and language models from an original point of view.
3. The proposed method performs very well in practice, as demonstrated in the experiments.
4. The paper showcases a real problem, provides an analysis to its cause, proposed a solution and provides experiments supporting the claims made and demonstrating superior performance of the proposed solution.


Weaknesses and questions:
1. First, I strongly recommend to change the terminology of "Cognitive Disorder". The term has little connection to the problem it describes, it's a bit far-fetched to borrow a term originally used to describe a mental health disorder and really adds nothing to the paper. A more technical term (dataset forgetting, degradation from distribution shift,...) would be much better.
2. The analysis on the causes of the performance degradation is done on the test set. However, the proposed solution is of course implemented during training. We obviously can't add a loss term during training and optimise it on the test set, but how much of this analysis holds when you consider the training set? I would expect the model to overfit to the training set, would there be inter-modal deviation between subset t and subset t-1 during training?
3. In table 3 alpha is set to 20. An analysis of the effect of alpha and to what extent the method is robust to the choice of alpha would be interesting to see. Of course, as alpha is increased the effect of "alignment" between models is increased, but I think the paper should explain how alpha was selected to be 20 and how sensitive this value is (e.g. how dramatically the results would change if a different, less ideal value would be used).
4. A drawback of the method is that it required to store the vision and language embeddings from model t-1 during training of model t (or re-evaluate them during step t). I'm not sure how much it would be applicable to training on huge scale datasets (e.g. LAION dataset).

**Summary Of The Paper:**

The paper addresses continual learning in joint vision and language model. Specifically, the paper analyses the causes for task/dataset forgetting in a continuous training regime from a geometric point of view. The analysis works as follows:

1. First, CLIP is trained on COCO to obtain a baseline CLIP0 model.
2. Next, Flicker 30K is divided into 5 subsets and the CLIP0 model is further trained sequentially on each of the subsets to obtain CLIP1-CLIP5 models. These models are also denoted as CLIP_ct models, where ct stands for continual training.
3. In addition, the authors train CLIP on the entire COCO and Flicker 30K datasets to obtain CLIP_jt, where jt stands for joint training, which serves as an upper bound to CLIP1-5 performance.
4. The authors measure the performance of the various models on Flicker and COCO test set, demonstrating a gradual degradation in performance of the CLIP_cp models on COCO tests set and an improvement on Flicker test set, which makes sense as the models are trained (gradually) on the Flicker dataset. They name this phenomenon "Cognitive Disorder".
5. Next, the paper considers the angles between all pairs of visual representations from different models, i.e.: the angle between the visual representation extracted using CLIP_t for a sample i and the visual representation extracted using CLIP_(t+1) for the same sample i. This analysis is done on the test set and denoted as Self Angle relationship Matrix (SAM). This measures to what extent the representations are rotated during continual training, with respect to themselves. They find that those angles change slowly, therefore argue to not be the reason for the performance degradation. The change in SAM is denoted as Intra-modal rotation.
6. The paper then turns to analyse how angles between pairs of visual embeddings from different samples change during continual learning. Specifically, they consider a matrix composed of the angles of representations from different samples i,j extracted from the same CLIP_t model. This matrix is denoted as Rotation Angle Matrix (RAM). This analysis is done for the textual embeddings as well.
7. The authors find that the RAM matrix changes more dramatically for visual embeddings than the angle between off-diagonal elements of the textual embeddings. The change in RAM is denoted as Inter-modal deviation.
8. From this analysis, the paper argues that the inter-modal deviation (i.e.: the fact that the two modalities are not synchronised in the amount of rotation between off-diagonal elements in different training steps) is the cause for the degradation in performance during continual learning (which is denoted Cognitive Disorder as I explained above).

Now, following this analysis, the paper proposes to alleviate "cognitive disorder" by aligning the amount of rotation of off-diagonal elements between the two modalities during training. This works as follows: first, given the trained model on sub dataset t , we consider the contrastive matrix M_t, i.e.: the matrix whose element in the i,j position is the similarity score of the i'th sample vision embedding and the j'th sample text embedding. The method add a KL divergence term between M_(t-1) to M_t to the loss. Samples (rows) for which the model in "timestamp" t-1 are not corrected classified are replaced in M_(t-1) by their values from M_t.

The paper presents two experiments to evaluate the performance of the proposed method. The first experiment follows the same setting as the analysis above and in the second experiment, the authors trained a CLIP0 model on COCO and Flickr datasets and gradually trained 8 additional models on 8 subsets of SBU-1M. The experiments demonstrate that using the proposed method, the resulting models only suffer from a small degradation of performance on the "baseline" dataset while performing well on the "new" gradually added sub-datasets, compared to the baseline CLIP0 model and even perform better on the "new" dataset than the baseline continual training model.





**Summary Of The Review:**

Strong paper in my opinion. The paper identifies a continual learning performance degradation as drawback of joint vision and language embedding models. It demonstrates this drawback by training a baseline model on COCO, "finetuning" it on Flick and evaluating on both Flickr and COCO test set, indicating a clear degradation on COCO. The paper analyses the cause for it and finds it to be Intra modal deviation, i.e.: misalignment between the amount of rotation of diagonal elements during training between the two modalities and designs a solution via adding a term to the loss to encourage intra modal alignment. The paper provides experiments that demonstrate the affect of the solution in mitigating the drawback. I mentioned a few weaknesses and question I would appreciate if the authors could address in their rebuttal.

---

> ### Author Response · Authors · 2022-11-11
> **Response to Reviewer 7TxY**
>
> We sincerely thank you for the valuable comments. We are encouraged to see that our work is recognized as novel,interesting and effective. We will respond to some of your suggestions and concerns point by point.
>
> **Q1: The terminology "Cognitive Disorder" has little connection to the problem the paper describes.**
> **A1:** Nice suggestion. We will replace it with another technical term in the revision version.
>
> **Q2: What will happen if you analysis the variation of inter-modal representation space using training set?**
> **A2:** Thanks for your question. In section 7.3, we add a more direct statistical analysis,inter-modal angle variation distribution. In the training phase t, we compare the change of angle distribution between modalities for the training samples retrieved correctly in the training phase t-1. The Figure 8(b) shows that, during the continual CLIP training, the inter-modal angle of the samples that were correctly retrieved in the past have apparent changes as the training phases go up. This shows that inter-modal deviation does exist in continual CLIP training. And the inter-modal spatial alignment (similarity performance) of the $CLIP_{ct}$ is affected by the inter-modal deviation.
>
> **Q3: How to choose a suitable hyper-parameter $\alpha$ during continual training?**
> **A3:** Thanks for raising this concern. In section 7.4.1, we add experiments to show that although different $\alpha$ affects the performance of the $CLIP_{Mod-X}$, different $\alpha$ does not significantly affect the effectiveness of the Mod-X framework. As the $\alpha$ increases, the model's ability to maintain the structure of representation space for old samples becomes stronger, while the ability to fit the current training samples decreases relatively. We empirically find that the initial choice of $\alpha$ mainly depends on the magnitude of loss $L_{InforNCE}$ and loss $L_{KL}$. The purpose of $\alpha$ is to ensure the balance of the effect of loss $L_{InforNCE}$ and loss $L_{KL}$ in the training process.
>
> **Q4: The Mod-X framework needs to save the embeddings of the current training sample on the past model during the training. Will this affect the training on large-scale datasets?**.
> **A4:** Thanks for raising a concern about Mod-X's training efficiency on large-scale datasets. During the continual CLIP training, the Mod-X framework only needs to extract the representation of the training samples in the current training step using the past model for the training of the current training step. Each training step only stores the representation of the current step of the old model. Therefore, there is no need to save these representations during continual training. And the requirements for storage space are not huge, and there will be no long-term storage space occupation. To illustrate the Mod-X framework is effective on large-scale datasets, in section 7.7, we add experiments to show that our Mod-X framework is still effective on the large-scale CC12M dataset. Due to time constraints, the validation of LAION will be done in the future.

---

> ### Author Response · Authors · 2022-12-03
> **Response to Reviewer 7TxY**
>
> Dear reviewer, we have tried to address your concerns in our earlier response. If you have any further questions or suggestions, we are very happy to discuss with you.

---

> > ### Comment · Reviewer_7TxY · 2022-12-07
> > **Response to authors**
> >
> > Thank you for your reply, you have addressed my concerns.

---

### Author Response · Authors · 2022-11-11
**General Response to All Reviewers**

We thank all the reviewers for their insightful and valuable comments! Overall, we are encouraged that they find that:

1.Exploring the feasibility of continual CLIP training is a **practical,interesting, and challenging problem** **(all reviewers)**.
2.The paper thoroughly analyzes the causes of retrieval performance drop during continual CLIP training, named intra-modal rotation and inter-modal deviation **(all reviewers)**.
3.The paper proposed an effective and practical solution that significantly alleviates the degradation of retrieval performance during continual CLIP training **(all reviewers)**.

However, we find some reviewers somewhat confused about the relationship between the inter-modal spatial variation proposed in our paper, the retrieval performance of the CLIP model, and our proposed Mod-X framework. For this reason, we reorganize the logical thinking of the paper here.

As the process summarized by reviewer 7TxY, first of all, we find that unlike continual self-supervised training based on single-modal images, the retrieval performance of the multi-modal CLIP model decreased significantly during continual training. By analyzing the variation of representation space during continual CLIP training, we find that the reason for the degradation of model retrieval performance is the shifting between the representation spaces of the two modal encoders. We disassemble this representation space shift into the intra-modal rotation and inter-modal deviation. Because of this, we propose the Mod-X framework. Detailed experiments demonstrate that our method indeed alleviates the CLIP's degradation in retrieval performance during continual training.

To better illustrate how the Mod-X framework mitigates performance degradation during continual CLIP training. We have added Section 7.3 in the Appendix to show how intra-modal spatial shift affects CLIP's retrieval performance and how our framework maintains an inter-modal representation space for the past retrieved correct samples during continual training.

Besides this, according to the reviewers' comments, we have revised the manuscript and added some sections in Appendix.
1.In section 7.3, we add experiments to show the relationship between contrastive matrix and intra-modal rotation and inter-modal deviation and Mod-X.
2.In section 7.3.1, we add experiments to show that inter-modal spatial shift still exists when continual training on a specific data domain.
3.In section 7.4.1, we add experiments to show that the choice of hyperparameter $\alpha$ does not significantly affect the effectiveness of the Mod-X framework.
4.In sections 7.7 and 7.8, we add experiments to show that our Mod-X framework is not only effective on the large-scale CC12M dataset but also significantly outperforms the baseline in "fine-tuning" on specific data domains (e-commerce dataset ECommerce-T2I).

Next, we address each reviewer's detailed concerns point by point. We hope we have addressed all of your concerns. Discussions are always open. Thank you!

---

> ### Author Response · Authors · 2022-12-03
> **General Response to All Reviewers**
>
> Dear reviewers, we have tried to address some concerns in our earlier response. If you have any further questions or suggestions, we are very happy to discuss with you.

---

### Decision · Program_Chairs · 2023-01-20

**Decision:**

Reject

**Justification For Why Not Higher Score:**

N/A

**Justification For Why Not Lower Score:**

N/A

**Metareview: Summary, Strengths And Weaknesses:**

This paper investigates the feasibility of training CLIP model continuously through streaming data and find the reason about cognitive disorder in continual CLIP training and proposes a new framework Mod-x to alleviate model's cognitive disorder. While the paper presents some interesting new ideas for continual learning in a new area, reviewers had some major concerns about the weak empirical studies, the lack of comparison with strong continual learning baselines (given this topic has been so much extensively studied), weak technical novelty,  etc. While the authors have attempted to address some review questions in their rebuttal, the manuscript requires a significant revision before it can be accepted in a top venue.